# Evaluating and Improving Subspace Inference in Bayesian Deep Learning

## Abstract

Bayesian neural networks incorporate Bayesian inference over model weights to account for uncertainty in weight estimation and predictions. Since full Bayesian inference methods are computationally expensive and suffer from high dimensionality, subspace inference has emerged as an appealing class of methods for approximate inference, where inference is restricted to a lower-dimensional weight subspace. Despite their benefits, existing subspace inference methods have notable pitfalls in terms of subspace construction, subspace evaluation, and inference efficiency. In this work, we conduct a comprehensive analysis of current subspace inference techniques and address all the aforementioned issues. First, we propose a block-averaging construction strategy that improves subspace quality by better resembling subspaces built from the full stochastic gradient descent trajectory. Second, to directly evaluate subspace quality, we propose novel metrics based on the Bayes factor and prior predictive, focusing on both goodness-of-fit and generalization abilities. Finally, we enhance inference within the subspace by leveraging importance sampling and quasi-Monte Carlo methods, significantly reducing computational overhead. Our experimental results demonstrate that the proposed methods not only improve computational efficiency but also achieve better accuracy and uncertainty quantification compared to existing subspace inference methods on CIFAR and UCI datasets.

## 1 Introduction

Traditional neural network models often rely on optimization methods to obtain point estimation of model weights without quantifying uncertainty. To address this, Bayesian neural networks (BNNs) incorporate Bayesian inference, providing a probabilistic interpretation of the model weights and predictions. Enabling uncertainty quantification not only improves model robustness (Gawlikowski et al., 2023), but also aids decision-making that are critical in high-risk areas such as medical image analysis (Lambert et al., 2024; Nair et al., 2020) and autonomous vehicles (Yang et al., 2023).

Despite their advantages, BNNs are challenged by the high dimensionality of the weight spaces, which can make training and inference computationally intensive and practically infeasible. Subspace inference (Izmailov et al., 2020; Garipov et al., 2018; Li et al., 2018) emerges as a compelling solution by projecting the weight space of a BNN into a low-dimensional subspace. This approach attempts to preserve the performance of the BNN while greatly reducing the computational overhead.

However, current subspace inference methods have several notable limitations. For instance, Li et al. (2018) proposes using random subspace to project low-dimensional weights into high-dimensional spaces, without considering how to select optimal projection directions. Izmailov et al. (2020) constructs the projection matrix based only on the tail trajectory of stochastic gradient descent (SGD), which can fail to capture the complete variability and structure of the entire trajectory. Moreover, existing metrics evaluate subspace quality only indirectly, relying on downstream task performance (e.g., log-likelihood or accuracy), without considering the interpretability of the subspace itself with respect to the training or testing data. To the best of our knowledge, there are currently no metrics designed to directly evaluate how well the subspace represents the data-relevant portions of the full parameter space. We emphasize that if the subspace does not adequately capture the high-quality points from the full space—those with lower loss or higher likelihood, any inference or uncertainty quantification derived from it may be unreliable.

**Our contributions.**    In this paper, we improve subspace inference in Bayesian deep learning in terms of subspace construction, subspace evaluation, and inference efficiency. We introduce novel subspace construction methods, called block-averaging (BA) techniques along the SGD trajectory. Unlike methods that focus only on the tail of the trajectory, our approach effectively captures the global features of the entire trajectory while maintaining the same cost as tail based methods. Furthermore, we introduce new evaluation metrics, such as the Bayes factor and prior predictive checks, to assess the quality of these subspaces—an aspect largely overlooked in previous work.

To further improve the efficiency of uncertainty quantification, we apply quasi-Monte Carlo and self-normalized importance sampling methods for inference the posterior predictive. We find that these strategies significantly reduce computational overhead in low-dimensional subspaces, while delivering prediction results that are comparable to more computationally intensive approaches.

**Related Works.**    Scaling Bayesian inference to high-dimensional models such as deep neural networks presents significant challenges. Some studies address this by constructing sparse subspaces in Bayesian neural networks, using sparsity-promoting priors (Molchanov et al., 2017; Deng et al., 2019; Ghosh et al., 2019) or applying removal-and-addition strategy (Li et al., 2024). Rather than zeroing out certain weights, some approaches fix parameter values and limit inference to a select subset, including methods that focus on the last layer (Kristiadi et al., 2020; Daxberger et al., 2021a) or on a carefully selected subset of model weights (Daxberger et al., 2021b). Besides Bayesian neural networks, research on deterministic neural networks has also demonstrated that deep networks can be effectively optimized within a low-dimensional subspace (Li et al., 2018; Gressmann et al., 2020; Wortsman et al., 2021; Jiang et al., 2022). Methods like SWA (Izmailov et al., 2018), TWA (Li et al., 2022a), and DLDR (Li et al., 2022b) further leverage the SGD trajectory to identify an optimal solution within the subspace. While these approaches demonstrate the potential for low-dimensional optimization, many existing methods either focus on specific layers or ignore the need for comprehensive evaluation metrics to assess subspace quality. Our work builds on this foundation by focusing on both subspace construction and evaluation, providing new tools for efficiently estimating uncertainty while ensuring high-quality inference within the subspace.

## 2 PRELIMINARIES

Denote a neural network by $f_w$, where $w$ denotes the weights. Given a training dataset $D$ consisting of input features $X$ and output labels $Y$, conventional neural network methods typically find the optimal weights $w^\star = \arg\min L(f_w(X), Y)$ with respect to a loss function $L$ using stochastic gradient descent (SGD). In Bayesian deep learning, we transform the loss function into a likelihood

$$\ell(w; X, Y) = \log p_w(Y \mid X) := -L(f_w(X), Y) \tag{1}$$

and study the posterior distribution $p(w \mid X, Y) \propto p(w)p(Y \mid X, w)$, where $p(w)$ is some prior distribution over the weights. We will write the likelihood as $p(D \mid w)$ for simplicity. Since Bayesian computational techniques suffer from the curse of dimensionality, subspace inference methods aim to construct a low dimensional posterior distribution that is (1) easier to estimate, and (2) still retains meaningful properties of $p(w \mid D)$.

Throughout this work we denote the high dimensional weight space by $\mathcal{W} \subseteq \mathbb{R}^d$ and a low dimension subspace with rank $k$ by $\mathcal{Z} \subset \mathcal{W}$. For example, Li et al. (2018); Izmailov et al. (2020) use linear subspaces $\mathcal{Z} = \{\hat{w} + Pz \mid z \in \mathbb{R}^k\}$ where vector $\hat{w} \in \mathbb{R}^d$ and projection matrix $P \in \mathbb{R}^{d \times k}$ are randomly generated.

Using this transformation and the original BNN weight likelihood $p_{\mathcal{W}}(D \mid w)$, there is an induced $\mathcal{Z}$-space likelihood $p_{\mathcal{Z}}(D \mid z) = p_{\mathcal{W}}(D \mid \hat{w} + Pz)$. After introducing a $\mathcal{Z}$-space prior distribution $p_{\mathcal{Z}}(z)$, there is an induced $\mathcal{Z}$-space posterior with an un-normalized density

$$p_{\mathcal{Z}}(z \mid D) \propto p_{\mathcal{Z}}(z) \, p_{\mathcal{Z}}(D \mid z). \tag{2}$$

Posterior predictive distributions provide uncertainty calibrations: for testing data $D'$, we would have

$$p_{\mathcal{W}}(D' \mid D) = \int_{\mathcal{W}} p_{\mathcal{W}}(D' \mid w)p_{\mathcal{W}}(w \mid D)\mathrm{d}w, \tag{3}$$

in the $\mathcal{W}$-space, and equation 3 is an integral over $\mathcal{W} \subseteq \mathbb{R}^d$. In subspace inference, one approximates equation 3 with

$$p_{\mathcal{Z}}(D' \mid D) = \int_{\mathbb{R}^k} p_{\mathcal{Z}}(D' \mid z) p_{\mathcal{Z}}(z \mid D) \mathrm{d}z, \tag{4}$$

which only requires an integration over the lower dimensional space. To approximate the integral in equation 4, a common approach is to obtain approximate samples from the induced posterior $p_{\mathcal{Z}}(z \mid D)$. This can be done using methods like Variational Inference (VI) (Blei et al., 2017) or Markov chain Monte Carlo (MCMC) methods such as Elliptical Slice Sampling (ESS) (Murray et al., 2010), Hamiltonian Monte Carlo (HMC) (Neal, 2012) and the No-U-Turn Sampler (NUTS) (Hoffman et al., 2014). With samples $z_1, z_2, \cdots, z_N \sim p_{\mathcal{Z}}(\cdot \mid D)$, the posterior predictive can be approximate with

$$p_{\mathcal{Z}}(D' \mid D) = \int_{\mathbb{R}^k} p_{\mathcal{Z}}(D' \mid z) p_{\mathcal{Z}}(z \mid D) \mathrm{d}z \approx \frac{1}{N} \sum_{i=1}^{N} p_{\mathcal{Z}}(D' \mid z_i). \tag{5}$$

## 3 Drawbacks of Existing Methods

In this section, we analyze the key limitations of existing subspace inference methods, particularly in subspace construction and posterior predictive estimation.

### 3.1 PCA-based Subspace Construction

To construct a subspace $\mathcal{Z} = \{\hat{w} + Pz \mid z \in \mathbb{R}^k\}$, one needs to learn $\hat{w}$ and $P$ from the data. Constructing subspaces based on Principal Component Analysis (PCA) (Garipov et al., 2018; Izmailov et al., 2020; Maddox et al., 2019) has proven to be effective in reducing dimensionality while preserving key information from the SGD trajectory. A simple strategy involves using the mean of the entire trajectory as $\hat{w}$, and then performing PCA on the matrix formed by the full trajectory (FT) to obtain projection matrix $P$. This is equivalent to applying singular value decomposition (SVD) on the centered matrix, where the trajectory is centered by subtracting $\hat{w}$. However, a significant challenge with this approach is the memory overhead. Full trajectory-based PCA (or SVD) requires storing and analyzing all training steps, which is infeasible in high-dimensional settings.

To address this, Izmailov et al. (2020) proposed a tail trajectory (TT) subspace construction method. In this approach, $\hat{w}$ is still constructed from the mean of the entire trajectory, but uses the deviations from the last $M$ points of the trajectory to $\hat{w}$ to construct the projection matrix $P$ via randomized SVD (Halko et al., 2011). While focusing on the tails reduces memory cost from $n$ to $M$, it introduces two key problems. First, using only the deviations from the tail leads to a deviation matrix that is not zero-centered, which makes the projection matrix constructed using SVD become dubious and no longer follows the interpretations of PCA. More importantly, using $M$ points from the tail of the trajectory can result in a subspace that only captures a small part of the whole trajectory. As a result, TT will underestimate variations in the trajectory and, even worse, lose the information on certain high-quality weights reflected in the full trajectory, as demonstrated by the heat maps of the induced likelihood in Figure 1.

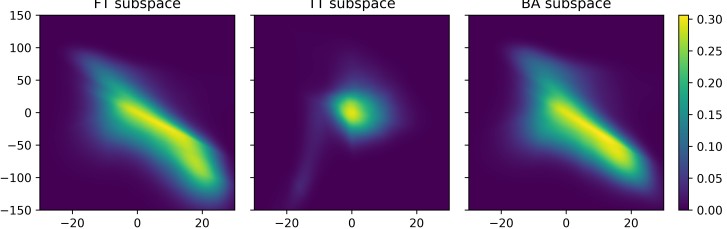

Figure 1: Heat maps of induced likelihood $p_{\mathcal{Z}}(D' \mid z)$ across full trajectory subspace, tail trajectory subspace, and our proposed block-averaging subspace with $M = 20$ on synthetic example.

### 3.2 Posterior Predictive via Bayesian Model Averaging

Existing methods for posterior predictive estimation, such as VI and MCMC, often incur significant computational costs. MCMC methods are flexible and asymptotically exact, allowing them to better

capture complex posterior distributions. However, they are computationally expensive due to the need for burn-in periods and the potential for low acceptance rates. Some MCMC methods like HMC and NUTS and VI require gradient evaluations, which greatly increase the computational burden.

Additionally, existing work has overlooked this correspondence between $p_{\mathcal{W}}(w)$ and $p_{\mathcal{Z}}(z)$. As a result of the transformation $w = \hat{w} + Pz$, a prior on $w$ on the full space uniquely determines an induced prior on $z$ after linear transformation, which should be used during subspace inference. A fixed prior for $w$ ensures the posterior reflects only the subspace properties, avoiding artifacts from varying priors. For instance, if the subspace center $\hat{w}$ shifts within the same subspace, the subspace $\mathcal{Z}$ remains unchanged, and the posterior structure does not vary. Instead, existing works manually choose the prior distribution for $z$.

Finally, the current practice of using the predictive performance in equation 5 to evaluate the quality of subspaces is indirect. We argue that a subspace evaluation step should be performed prior to subspace inference, as a poorly constructed subspace is likely to result in suboptimal performance, regardless of the inference method.

## 4 OUR METHODS

### 4.1 BLOCK-AVERAGING SUBSPACE CONSTRUCTION

We propose block-averaging (BA) subspace construction. It can better capture the variations in the full SGD trajectory than the tail trajectory. BA can be viewed as a structured downsampling of the full trajectory, providing a similar subspace as shown in Figure 1, while reducing computational and memory costs. Specifically, BA partitions the trajectory into $M$ equidistant blocks and constructs a $M \times d$ matrix $\overline{W}$ consisting of block centers. We propose an *online* implementation for BA construction in Algorithm 1. Notably, it has the same algorithmic complexity and memory cost as the TT construction. Figure 2 illustrates and contrasts the subspaces constructed using the full trajectory, the tail of the trajectory, and our block-averaging strategy.

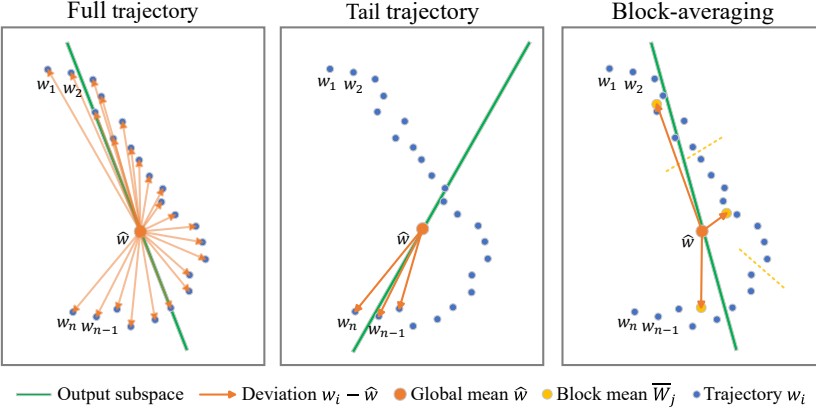

Figure 2: Visualization of different subspace construction methods.

The block-averaging strategy captures the directions of the largest variations across the entire trajectory, rather than focusing only on its tail. As a result, the subspace includes more representative and diverse points from the $\mathcal{W}$-space, enhancing the quality of the subspace for uncertainty evaluation.

Our block-averaging subspace construction achieves comparable uncertainty calibration to the full trajectory while significantly reducing the computational cost. We use the synthetic data regression example from Izmailov et al. (2020) as a running example to illustrate the advantages of our methods. We construct subspaces using a $n = 1000$ point trajectory and memory cost $M = 20$. In Table 1, we measure subspace distances via the angular distance between their first principal components. The full trajectory and block-averaging subspaces are similar, while the tail trajectory subspace is nearly orthogonal to the full trajectory. As a result, the BA subspace contains more high likelihood (low loss) weights than TT subspace. This will be further illustrated with Figure 5 in the real-world data experiments on UCI datasets.

---

**Algorithm 1** Block-Averaging Subspace Construction

---

**Input**: loss function $\mathcal{L}$, pre-trained initial weights $w_0$; learning rate $\eta$; number of SGD iterations $n$; number of clocks $M$;
**Initialization**: initialize global mean $\hat{w} \leftarrow w_0$, block counter $c_1 = 0, \cdots, c_M = 0$
**for** $i \in [1, 2, \cdots, n]$ **do**
    Update $w_i \leftarrow w_{i-1} - \eta \nabla_w \mathcal{L}(w_{i-1})$                  ▷ SGD step
    Update $\hat{w} \leftarrow (i\hat{w} + w_i)/(i+1)$             ▷ Update global mean
    Get current block order $j \leftarrow \lfloor (i-1)M/n \rfloor + 1$
    Update the $j$-th row of matrix $W$ with $W_j \leftarrow (c_j W_j + w_i)/(c_j + 1)$    ▷ Update block mean
    Update counter $c_j \leftarrow c_j + 1$
**end for**
Update $W \leftarrow W - [\hat{w}^\top, \cdots, \hat{w}^\top]^\top$              ▷ Center the weight matrix
$U, \Sigma, V^\top \leftarrow \text{SVD}(W)$
**return** $\hat{w}, P = V^\top \Sigma/(\sqrt{M-1})$

---

Table 1: Subspace angles (degrees) between different methods. 0 indicates identical subspaces and 90 reflects orthogonal subspaces. Values are reported as mean±sd.

| Method | Full trajectory | Tail trajectory | Block-averaging |
|---|---|---|---|
| Full trajectory | 0 | 87.184±2.237 | 3.227±4.604 |
| Tail trajectory | 87.184±2.237 | 0 | 87.213±2.389 |
| Block-averaging | 3.227±4.604 | 87.213±2.389 | 0 |

### 4.2 Two Subspace Quality Evaluations: Bayes Factor and Prior Predictive

There have been no quantitative evaluation methods to assess the quality of subspaces. After subspace construction, current works skip subspace evaluation and directly start making predictions using the posterior predictive distribution. Here, we propose two principled ways for subspace evaluation.

In Bayesian statistics, each model is a parametric family of probability distributions we use to explain the observed data $D$. In the context of BNN, a model is a neural network and its parameter is a set of weights $w \in \mathcal{W}$. In subspace inference, each subspace is a sub-model restricting the weights to $\mathcal{Z} = \{\hat{w} + Pz \mid z \in \mathbb{R}^k\}$. In $p_\mathcal{W}(w \mid D) = p_\mathcal{W}(w)p_\mathcal{W}(D \mid w)/p_\mathcal{W}(D)$, the normalizing constant $p_\mathcal{W}(D)$ is the *evidence* of the full model in the context of Bayesian model selection.

**Definition 1** (Subspace evidence). For a subspace $\mathcal{Z}$ and observed dataset $D$, we define subspace evidence as the marginal likelihood of $D$ restricted to this subspace, i.e.

$$p(D \mid \mathcal{Z}) = \int_{w \in \mathcal{Z}} p_\mathcal{W}(D \mid w)p_\mathcal{W}(w)\mathrm{d}w = \int_{z \in \mathbb{R}^k} p_\mathcal{Z}(D \mid z)p_\mathcal{Z}(z)\mathrm{d}z. \tag{6}$$

In the spirit of *Bayesian model comparison*, we can use the ratio of marginal likelihoods to choose between two competing hypotheses (which are subspaces in the context of BNN). This ratio is the Bayes factor (Kass & Raftery, 1995).

**Definition 2** (Bayes factor for subspaces). With two subspaces $\mathcal{Z}_1$ and $\mathcal{Z}_2$, their Bayes factor in favor of $\mathcal{Z}_1$ concerning observed data $D$ is

$$\text{BF}_{1,2} = \frac{p(D \mid \mathcal{Z}_1)}{p(D \mid \mathcal{Z}_2)}.$$

Jeffery's scale of evidence (Kass & Raftery, 1995; Wasserman, 2000; Berger, 2003) gives an interpretation for Bayes factors: With $\text{BF}_{1,2} > 10$, there is strong evidence favoring subspace $\mathcal{Z}_1$ and $\text{BF}_{1,2} > \sqrt{10} \approx 3.2$ gives substantial evidence for $\mathcal{Z}_1$. Similarly $\text{BF}_{1,2} < 0.1$ or $\text{BF}_{1,2} < 0.32$ gives strong / substantial evidence for choosing $\mathcal{Z}_2$.

Continuing with the examinations in Table 1, we compare the TT and BA subspaces against the full trajectory construction in Figure 3 (Left) using Bayes factors. Using Jeffery's scale for interpreting Bayes factors, there is strong evidence for choosing the subspace constructed from the full trajectory against the tail construction with $M < 5$ points. On the other hand, subspaces from the full and the

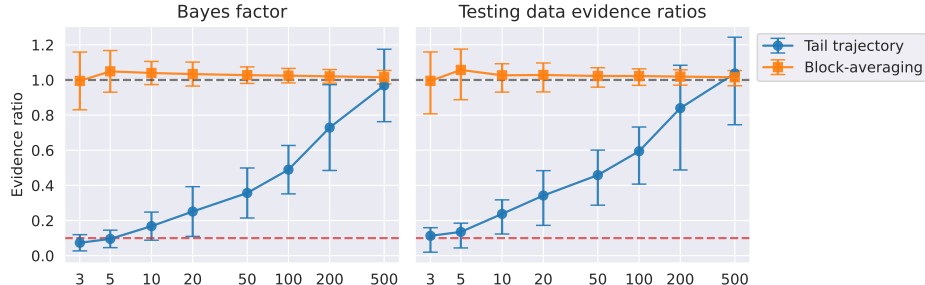

Figure 3: Evidence ratios for different subspace construction methods using $M = 3, 5, 10, 20, 50, 100, 200, 500$ points. Full trajectory length is $n = 1000$. Left: Bayes factor against full trajectory subspace. Right: testing data evidence ratios against full trajectory subspace. Error bars indicate the mean±sd.

BA trajectories are comparable. In addition, high deviation values of the Bayes factors show that the quality of the tail construction is unstable across SGD trajectories.

One can also determine the quality of subspaces qualities using a hold-out data $D'$.

**Definition 3** (subspace evidence on test data).

$$p(D' \mid \mathcal{Z}) = \int_{w \in \mathcal{Z}} p_{\mathcal{W}}(D' \mid w) p_{\mathcal{W}}(w)\mathrm{d}w = \int_{z \in \mathbb{R}^k} p_{\mathcal{Z}}(D' \mid z) p_{\mathcal{Z}}(z)\mathrm{d}z. \qquad (7)$$

In Def. 3, higher test evidence $p(D' \mid \mathcal{Z})$ indicates that *on average* weights in the $\mathcal{Z}$ subspace can better explain the test data. We highlight that equation 7 differs from the subspace evidence in equation 6, because in equation 7, the subspace $\mathcal{Z}$ is trained using observed data $D$; it also differs from the posterior predictive in equation 4. Indeed, while one might interpret equation 7 as the *prior predictive distribution* on subspace $\mathcal{Z}$, using the term *evidence on test* can reflect the fact the subspace itself is constructed using training data. When $\mathcal{Z}$ is constructed using training data $D$, equation 7 reflects to what extent the subspace constructed using training data can fit the testing data.

The evidence ratio on testing data

$$\mathrm{ER}_{1,2} = \frac{p(D' \mid \mathcal{Z}_1)}{p(D' \mid \mathcal{Z}_2)},$$

can also compare subspace qualities, with a larger ratio indicating a stronger preference for choosing $\mathcal{Z}_1$ over $\mathcal{Z}_2$. Using the evidence ratio on testing data, we confirm that the subspaces constructed from the BA trajectory outperform those from TT in Figure 3 (Right).

### 4.3 USING IMPORTANCE SAMPLING AND QUASI-MONTE CARLO FOR POSTERIOR PREDICTIVE

In contrast to MCMC and VI methods reviewed in Section 2, we propose several importance sampling (IS) based methods to provide predictions with uncertainty quantification for BNN, which is motivated by the following observation. Weights from the trajectory used to train $\mathcal{Z}$ have an empirical mean of 0 and an empirical covariance of $I_k$ after dimension reduction. This fact suggests that the induced $\mathcal{Z}$ space posterior distribution could be similar to a standard multivariate Gaussian, making importance sampling a compelling inference technique to study $p_{\mathcal{Z}}(z \mid D)$.

Recall from equation 4 that the subspace predictive posterior on testing data $D'$ is $p_{\mathcal{Z}}(D' \mid D) = (p_{\mathcal{Z}}(D))^{-1} \int_{z \in \mathbb{R}^k} p_{\mathcal{Z}}(D' \mid z) p_{\mathcal{Z}}(D \mid z) p_{\mathcal{Z}}(z)\mathrm{d}z$, which follows from Bayes rule. Using some proposal density $q(z)$ and a change of basis from $p_{\mathcal{Z}}$ to $q$, we have

$$p_{\mathcal{Z}}(D' \mid D) = \frac{\int_{\mathcal{Z}} p_{\mathcal{Z}}(D' \mid z) p_{\mathcal{Z}}(D \mid z) \frac{p_{\mathcal{Z}}(z)}{q(z)} q(z)\mathrm{d}z}{\int_{\mathcal{Z}} p_{\mathcal{Z}}(D \mid z) \frac{p_{\mathcal{Z}}(z)}{q(z)} q(z)\mathrm{d}z} = \frac{\mathbb{E}_{Z \sim q}\left[p_{\mathcal{Z}}(D', D \mid Z) p_{\mathcal{Z}}(Z)/q(Z)\right]}{\mathbb{E}_{Z \sim q}\left[p_{\mathcal{Z}}(D \mid Z) p_{\mathcal{Z}}(Z)/q(Z)\right]} \qquad (8)$$

A natural self-normalized importance sampling (SNIS) estimator for equation 8 is

$$\widehat{p}_{\mathrm{IS}}(N, q; D, D') = \frac{\sum_{i=1}^{N} p_{\mathcal{Z}}(D', D \mid Z_i) p_{\mathcal{Z}}(Z_i)/q(Z_i)}{\sum_{i=1}^{N} p_{\mathcal{Z}}(D \mid Z_i) p_{\mathcal{Z}}(Z_i)/q(Z_i)}, \qquad (9)$$

Table 2: RMSE of posterior predictive estimations in different subspaces. The cost is measured by the number of forward passes through the model on the training set.

| Method | Full trajectory | | Tail trajectory | | Block-averaging | |
|---|---|---|---|---|---|---|
| | RMSE | Cost | RMSE | Cost | RMSE | Cost |
| ESS | 0.0091 | 6716±119.9 | 0.0110 | 5630±104.5 | 0.0091 | 6663±103.7 |
| VI | 0.0488 | 2000 | 0.0606 | 2000 | 0.0479 | 2000 |
| SNIS ($N = 256$) | 0.0137 | 256 | 0.0102 | 256 | 0.0141 | 256 |
| SNIS ($N = 1024$) | 0.0064 | 1024 | 0.0052 | 1024 | 0.0065 | 1024 |
| RQMC-IS ($N = 256$) | 0.0103 | 256 | 0.0031 | 256 | 0.0092 | 256 |
| RQMC-IS ($N = 1024$) | **0.0026** | 1024 | **0.0006** | 1024 | **0.0028** | 1024 |

where $Z_1, \cdots, Z_n$ are $N$ independent and identically distributed (iid) samples from $q$. Lemma 1 shows that the SNIS estimator consistently estimates the posterior predictive as the number of IS samples increases.

**Lemma 1.** *Under the Assumption 3, the equation 9 is a consistent estimator of the posterior predictive function, i.e.*

$$\mathbb{P}\left(\lim_{N \to \infty} \widehat{p}_{\mathrm{IS}}(N, q; D, D') = p_{\mathcal{Z}}(D' \mid D)\right) = 1. \tag{10}$$

*Proof.* This follows from the strong law of large numbers. See (Owen, 2013, Theorem 9.2). □

In practice, the intrinsic dimensionality of $\mathcal{Z}$ is chosen to be small (e.g., $k \leq 5$). This motivates us to use randomized quasi-Monte Carlo (RQMC) methods (Owen, 1997a; L'Ecuyer, 2018) to further improve the SNIS estimator in equation 9. Instead of using iid samples from the distribution $q$, RQMC methods generate correlated and low-discrepancy sequences within a $k$-dimensional unit hypercube. These sequences are then mapped to pseudo-samples that follow the distribution $q$ by applying $F_q^{-1}$, the inverse of the cumulative distribution function (CDF) of $q$. Specifically, we first generate a low-discrepancy sequence of length $N$ (e.g., scrambled digital nets (Owen, 1997b)) $\{U_1, \cdots, U_N\}$ from the $k$-dimensional unit hypercube $[0, 1)^k$. This sequence is transformed using the inverse CDF transformation $Z_i = F_q^{-1}(U_i)$. Finally, replacing $Z_i$ in equation 9 with these low-discrepancy samples results in the RQMC estimator

$$\widehat{p}_{\mathrm{RQMC}}(N, q; D, D') = \frac{\sum_{i=1}^{N} p_{\mathcal{Z}}(D', D \mid F_q^{-1}(U_i)) p_{\mathcal{Z}}(F_q^{-1}(U_i)) / q(F_q^{-1}(U_i))}{\sum_{i=1}^{N} p_{\mathcal{Z}}(D \mid F_q^{-1}(U_i)) p_{\mathcal{Z}}(F_q^{-1}(U_i)) / q(F_q^{-1}(U_i))}. \tag{11}$$

RQMC significantly reduces the root mean squared error (RMSE) of estimators, which is particularly useful in low-dimensional integration tasks (L'Ecuyer, 2018). A key advantage of RQMC is its faster convergence rate of $\mathcal{O}(N^{-1+\epsilon})$, compared to the standard Monte Carlo rate of $\mathcal{O}(N^{-1/2})$. This improvement is formalized in the following theorem, with the detailed proof and assumptions provided in Appendix A.

**Theorem 2.** *Under the Assumption 3 and 4, the RMSE for the RQMC-IS estimator satisfies*

$$\sqrt{\mathbb{E}\left[\left(\widehat{p}_{\mathrm{RQMC}}(N, q; D, D') - p_{\mathcal{Z}}(D' \mid D)\right)^2\right]} = \mathcal{O}(N^{-1+\epsilon}) \tag{12}$$

*for arbitrarily small $\epsilon > 0$.*

In Table 2 we compare the RMSE and computational cost of different computational approaches to evaluate the posterior predictive on a test data set, where the computational cost is measured by the number of evaluations of the induced likelihood $p_{\mathcal{Z}}(D \mid z)$. The results show that the RMSE achieved by the RQMC-IS method with $N = 1024$ is superior to the ESS and VI methods, while also requiring significantly less computational resources. Compared to the SNIS method, which is based on iid samples and has a convergence rate of $\mathcal{O}(N^{-1/2})$, RQMC-IS also exhibits a faster convergence rate.

## 5 EXPERIMENTS

We conducted comprehensive experiments to illustrate the performance of the proposed subspace construction, subspace evaluation, and inference methods. In Section 5.1, we present results on

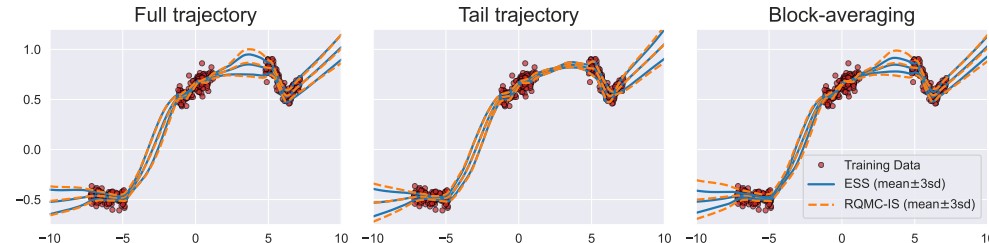

Figure 4: Visualizing uncertainty using posterior predictive: the full trajectory (FT) and block-averaging (BA) subspace reflect higher uncertainty in data-sparse regions and higher confidence in data-rich regions, while the tail trajectory (TT) tends to be overconfident.

Table 3: Bayes factors and testing data evidence ratios on UCI dataset (tail trajectory subspace against block-averaging subspace).

(a) Small UCI Regression Datasets

|  | boston | concrete | energy | naval | yacht |
|---|---|---|---|---|---|
| Bayes factor | $0.123 \pm 0.031$ | $0.340 \pm 0.244$ | $0.214 \pm 0.255$ | $0.018 \pm 0.020$ | $0.335 \pm 0.705$ |
| Evidence ratios | $0.157 \pm 0.052$ | $0.545 \pm 0.196$ | $0.291 \pm 0.212$ | $0.140 \pm 0.112$ | $0.199 \pm 0.091$ |

(b) Large UCI Regression Datasets

|  | elevators | protein | pol | keggD | keggU | skillcraft |
|---|---|---|---|---|---|---|
| Bayes factor | $0.215 \pm 0.121$ | $0.091 \pm 0.096$ | $0.178 \pm 0.077$ | $0.269 \pm 0.223$ | $0.214 \pm 0.252$ | $0.474 \pm 0.263$ |
| Evidence ratios | $0.544 \pm 0.184$ | $0.537 \pm 0.238$ | $0.205 \pm 0.089$ | $0.359 \pm 0.257$ | $0.218 \pm 0.288$ | $0.608 \pm 0.497$ |

synthetic example, showing that our method provides reliable posterior predictive distributions. Section 5.2 highlights that the proposed BA subspace consistently achieves higher Bayes factors and evidence ratios compared to the TT subspace across both small- and large-scale UCI regression tasks. Additionally, the RQMC-IS method delivers reliable predictions while requiring approximately half the computational budget compared to other methods. Finally, in Section 5.3, we evaluate our method's performance on CIFAR image classification tasks, where the BA-based subspace achieves higher test accuracy. We also demonstrate BA subspace's better performance on noisy data and out-of-distribution detection. See Appendix B, C and D for detailed experimental setup.

## 5.1 CASE STUDY: UNCERTAINTY QUANTIFICATION DURING REGRESSION

In this study, We demonstrate how subspace inference methods incorporate uncertainty into their predictions. BA-RQMC outperforms existing methods in the following senses: (1) unlike full trajectory and BA subspaces, the TT subspace is overly confident about extrapolation in predicting $y$ for unobserved $x$, and (2) the RQMC-IS method reduces the computational cost to about 15% compared to ESS and has better predictive accuracy.

Our experiments use the same synthetic data and neural network structure $f_w$ following Izmailov et al. (2020). Figure 4 shows the posterior predictive results for the full trajectory, TT, and our proposed BA subspace construction methods under $M = 20$. Compared to the TT subspace, the BA subspace reflects higher uncertainty in data-sparse regions and higher confidence in data-rich regions. The results are very close to those obtained using the FT, but our method offers a significant reduction in the computational cost when constructing the subspace (See the cost columns in Table 2). In Figure 6 (See Appendix B), we compare the likelihood heatmaps across different subspaces for both the training and testing data, as well as the posterior for the training data. The BA subspace contains more 'high-likelihood' sample points and a broader posterior high-density region, providing a better representation of the effective parameters in the full space $\mathcal{W}$ and offering a more informed assessment of uncertainty.

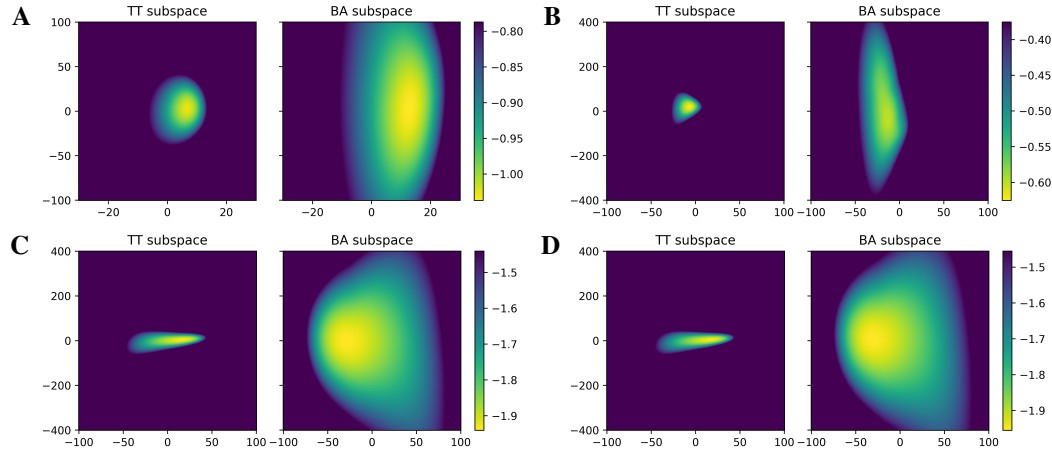

Figure 5: Heat maps of induced loss on different subspaces and datasets. **A.** Evaluated on `boston` training data. **B.** Evaluated on `boston` testing data. **C.** Evaluated on `keggundirected` training data. **D.** Evaluated on `keggundirected` testing data. The BA subspaces capture more low-loss points compared to TT subspaces.

Table 4: Test log-likelihood on UCI-Small datasets.

| Dataset | TT (ESS) | BA (ESS) | TT (NUTS) | BA (NUTS) | TT (VI) | BA (VI) | TT (RQMC) | BA (RQMC) |
|---|---|---|---|---|---|---|---|---|
| boston | -2.690±0.287 | -2.692±0.285 | -2.663±0.257 | **-2.657±0.210** | -2.692±0.292 | -2.654±0.271 | -2.690±0.296 | -2.700±0.289 |
| concrete | **-3.079±0.140** | **-3.080±0.140** | -3.115±0.116 | -3.102±0.130 | **-3.081±0.132** | **-3.083±0.126** | **-3.080±0.141** | **-3.082±0.140** |
| energy | -1.397±0.183 | **-1.396±0.187** | -1.473±0.193 | -1.487±0.168 | -1.403±0.176 | -1.439±0.166 | **-1.396±0.184** | **-1.394±0.184** |
| naval | 5.492±0.286 | **5.559±0.272** | -0.825±1.994 | 0.195±1.904 | 5.510±0.257 | 5.531±0.293 | 5.451±0.246 | **5.549±0.271** |
| yacht | -2.128±0.183 | -2.127±0.183 | -2.543±0.401 | -2.779±0.546 | -2.141±0.176 | -2.175±0.150 | -2.123±0.175 | **-2.103±0.175** |

## 5.2 UCI REGRESSION

Next, we compare our subspace inference methods on UCI regression tasks. For 5 small-scale datasets (`boston`, `concrete`, `energy`, `naval`, and `yacht`), we use the set-up proposed by Bui et al. (2016) and employ a small fully connected neural network. For larger-scale regression tasks, we experiment with 6 large datasets (`elevators`, `protein`, `pol`, `keggdirected`, `keggundirected`, and `skillcraft`) using a larger fully connected neural network, following the set-up of Wilson et al. (2016) and Izmailov et al. (2020). Tables 3a and 3b present the Bayes factors and evidence ratios, showing substantial evidence in favor of the BA subspace over the TT subspace. The detailed experimental setup is presented in Appendix C.

Figures 5-A and 5-B compare the loss on the training and testing data from `boston`, and Figures 5-C and 5-D further present the loss on the training and testing data from `keggundirected`. Since the loss is equivalent to negative log-likelihood in our settings, We observe that (1) the BA subspaces contain more 'low-loss' or 'high-likelihood' points, reflecting higher subspace quality, (2) the subspace origin (which corresponds to the center of the SGD trajectory $\hat{w}$ in the original weight space $\mathcal{W}$) does not necessarily maximize the likelihood in $\mathcal{Z}$.

For posterior predictive checks, we apply our proposed RQMC-IS method and compare it with ESS, NUTS, and VI. Tables 4 and 5 present the test log-likelihoods for UCI-Small and UCI-Large datasets, respectively, where our method achieves higher log-likelihoods on most datasets. Moreover, Table 10 demonstrates that our method has a lower computational cost. Additional results, including test RMSEs and test calibration results, are reported in Appendix C.

## 5.3 IMAGE CLASSIFICATION

We conduct classification experiments on the `CIFAR` datasets (Krizhevsky, 2009) using VGG-16 (Simonyan & Zisserman, 2014) and PreResNet164 (He et al., 2016), using the same setups with Maddox et al. (2019). First, we report the model evidence ratios in Table 14 (see Appendix D for details), which show substantial evidence in favor of the BA subspace over the TT subspace. For

Table 5: Test log-likelihood on UCI-Large datasets.

| Dataset | TT (ESS) | BA (ESS) | TT (NUTS) | BA (NUTS) | TT (VI) | BA (VI) | TT (RQMC) | BA (RQMC) |
|---|---|---|---|---|---|---|---|---|
| elevators | 1.009±0.024 | 1.009±0.025 | 0.264±0.342 | 0.286±0.244 | 1.008±0.026 | 1.008±0.027 | **1.013±0.022** | **1.011±0.023** |
| protein | -0.691±0.016 | **-0.684±0.016** | -1.224±0.070 | -1.182±0.114 | -0.694±0.015 | -0.697±0.015 | -0.689±0.015 | **-0.683±0.015** |
| pol | **-4.346±0.065** | -4.356±0.072 | -5.793±0.207 | -5.724±0.326 | -4.453±0.039 | -4.464±0.041 | -4.350±0.067 | -4.346±0.063 |
| keggD | 0.688±0.036 | 0.687±0.042 | 0.527±0.149 | 0.537±0.140 | 0.588±0.354 | 0.564±0.429 | 0.691±0.035 | **0.694±0.037** |
| keggU | 0.896±0.161 | 0.890±0.154 | 0.799±0.112 | 0.814±0.125 | 0.865±0.153 | 0.845±0.135 | **0.905±0.162** | 0.892±0.154 |
| skillcraft | -0.029±0.037 | -0.028±0.035 | -0.374±0.262 | -0.324±0.199 | **-0.021±0.041** | -0.022±0.042 | -0.020±0.038 | -0.021±0.038 |

predictive performance, we present classification accuracy in Table 6 and the corresponding negative log-likelihood in Table 15. The BA-based subspace, when combined with VI and RQMC methods, achieves higher accuracy. Due to the high dimensionality of weights, constructing FT subspace is computationally infeasible and thus omitted from our evaluations.

To further evaluate the generalization ability, we test robustness on the corrupted CIFAR datasets (Hendrycks & Dietterich, 2019), which contain various types of noise. As shown in Table 7, although classification accuracy declines as the severity level increases, the BA-based method consistently outperforms TT in terms of accuracy.

Additionally, we use out-of-distribution (OOD) data to demonstrate and compare the OOD detection performance of subspace inference methods. Specifically, we train the models on CIFAR and check whether the model can distinguish between CIFAR samples and SVHN samples (Netzer et al., 2011). The accuracy of each algorithm is evaluated using the area under the ROC curve (AUC), as shown in Table 17. All methods achieve over 75% AUROC, demonstrating the OOD detection ability of subspace methods in general. We find that the BA construction slightly outperforms the TT construction.

Table 6: Classification accuracy (ACC(%)) on CIFAR datasets.

| Models | TT (ESS) | BA (ESS) | TT (VI) | BA (VI) | TT (RQMC) | BA (RQMC) |
|---|---|---|---|---|---|---|
| VGG-16 on CIFAR10 | 91.98±0.43 | 91.92±0.40 | 91.80±0.42 | **92.00±0.44** | 91.76±0.37 | 91.94±0.51 |
| PreResNet164 on CIFAR10 | 94.99±0.17 | 95.08±0.11 | 94.96±0.15 | **95.13±0.11** | 95.05±0.12 | 94.92±0.06 |
| VGG-16 on CIFAR100 | **68.32±0.47** | 68.18±0.42 | 68.07±0.47 | 68.17±0.52 | 68.19±0.58 | **68.33±0.49** |
| PreResNet164 on CIFAR100 | 76.99±0.03 | 77.06±0.15 | 76.94±0.14 | 77.14±0.27 | 76.82±0.19 | **77.30±0.35** |

Table 7: Classification accuracy (ACC(%)) on corrupted CIFAR datasets using PreResNet164.

| Severity | TT (ESS) | BA (ESS) | TT (VI) | BA (VI) | TT (RQMC) | BA (RQMC) |
|---|---|---|---|---|---|---|
| 1 | 94.47±0.17 | **94.56±0.15** | 94.15±0.24 | **94.58±0.07** | 94.24±0.10 | 94.51±0.04 |
| 2 | 93.18±0.21 | 93.26±0.21 | 92.82±0.38 | **93.45±0.12** | 92.66±0.20 | 93.30±0.08 |
| 3 | 91.36±0.27 | 91.28±0.31 | 90.70±0.49 | **91.57±0.20** | 90.52±0.27 | 91.29±0.12 |
| 4 | 87.98±0.40 | 87.77±0.65 | 87.09±0.83 | **88.25±0.42** | 86.86±0.45 | 87.86±0.52 |
| 5 | 72.42±0.87 | 71.36±1.95 | 70.70±1.71 | **72.62±1.29** | 69.65±1.94 | 72.02±1.73 |

## 6 CONCLUSION

In this work, we have addressed several key limitations in subspace inference for Bayesian deep learning, proposing novel strategies to improve the construction, evaluation, and efficiency of these methods. Our BA subspace construction results in subspaces with more high likelihood, low loss weights and therefore better predictions and uncertainty quantification. Additionally, by introducing new subspace metrics based on the Bayes factor and prior predictive checks, we have provided a more comprehensive evaluation framework for subspace quality. These metrics focus on both the goodness-of-fit and generalization capabilities of the subspace, offering a more reliable means to assess and compare different subspace inference techniques.

Our experiments on benchmark datasets, including CIFAR and UCI, further highlight the benefits of our approach. In particular, our proposed RQMC-IS method achieves comparable performance to other techniques while maintaining a lower computational cost. This efficiency makes our method particularly suitable for large-scale datasets or complex networks, where computational resources might be a critical concern. In conclusion, the methods presented in this work offer a scalable solution for Bayesian deep learning by improving subspace construction and inference techniques.

## REPRODUCIBILITY STATEMENT

To ensure the reproducibility of our work, we have provided detailed assumptions and proof of Theorem 2 in Appendix A. The detailed experimental setup and results for each type of experiment are provided in Appendices B, C, and D. The source code is available in the Supplementary Material.

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

## A    DETAILS OF THEOREM 2

To establish the Theorem 2, we begin by detailing two critical assumptions.

**Assumption 3.** *The induced likelihood $p_{\mathcal{Z}}(D' \mid Z)$ is bounded by a constant $M_0$ for all $Z \in \mathcal{Z}$, and the proposal distribution $q$ satisfied $q(Z) > 0$ whenever $p_{\mathcal{Z}}(D \mid Z)p_{\mathcal{Z}}(z) > 0$.*

**Assumption 4.** *Define the functions $G_1(Z)$ and $G_2(Z)$ as*

$$G_1(Z) = p_{\mathcal{Z}}(D', D \mid Z)p_{\mathcal{Z}}(Z)/q(Z), \tag{13}$$

$$G_2(Z) = p_{\mathcal{Z}}(D \mid Z)p_{\mathcal{Z}}(Z)/q(Z), \tag{14}$$

*then $G_1(Z)$ and $G_2(Z)$ are subject to the boundary growth condition (He et al., 2023), i.e., for arbitrarily small $B_i > 0$, there exists a constant $B > 0$ such that*

$$|(D_v G_j)(Z)| \leq B \prod_{i=1}^{k} [\min(U^i, 1 - U^i)]^{-B_i}, \quad j \in \{1, 2\} \tag{15}$$

*holds for any $v \subset \{1, 2, \cdots, k\}$, where $(D_v G_j)(Z)$ denote the mixed partial derivative of $G_j$ with respect to each $i$-th element of $Z$ for all $i \in v$, and $U = (U^1, \cdots, U^k) = F_q(Z) \in (0, 1)^k$ where $F_q$ is the CDF of proposal distribution $q$.*

Next, we give the proof of Theorem 2.

**Theorem 2.** *Under the Assumption 3 and 4, the RMSE for the RQMC-IS estimator satisfies*

$$\sqrt{\mathbb{E}\left[\left(\widehat{p}_{\mathrm{RQMC}}(N, q; D, D') - p_{\mathcal{Z}}(D' \mid D)\right)^2\right]} = \mathcal{O}(N^{-1+\epsilon}) \tag{12}$$

*for arbitrarily small $\epsilon > 0$.*

*Proof.* We first denote the integrals as

$$I_1 = \int_{\mathcal{Z}} G_1(Z)q(Z)\mathrm{d}Z, \quad I_2 = \int_{\mathcal{Z}} G_2(Z)q(Z)\mathrm{d}Z,$$

and their unbiased estimators as

$$\hat{I}_{1,N} = \frac{1}{N}\sum_{i=1}^{N} G_1(F_q^{-1}(U_i)), \quad \hat{I}_{2,N} = \frac{1}{N}\sum_{i=1}^{N} G_2(F_q^{-1}(U_i)),$$

According to Scheichl et al. (2017), the RMSE of $\hat{p}_N(D' \mid D)$ can be bounded using the triangle inequality

$$
\sqrt{\mathbb{E}\left[(\hat{p}_N(D' \mid D) - p_{\mathcal{Z}}(D' \mid D))^2\right]} = \sqrt{\mathbb{E}\left[\left(\frac{\hat{I}_{1,N}}{\hat{I}_{2,N}} - \frac{I_1}{I_2}\right)^2\right]}
$$

$$
\leq \sqrt{\frac{2}{I_2^2}\left(\mathbb{E}\left[(\hat{I}_{1,N} - I_1)^2\right] + \mathbb{E}\left[\left(\frac{\hat{I}_{1,N}}{\hat{I}_{2,N}}\right)^2 (\hat{I}_{2,N} - I_2)^2\right]\right)}
$$

$$
\leq \frac{\sqrt{2}}{I_2}\sqrt{\mathbb{E}\left[(\hat{I}_{1,N} - I_1)^2\right]} + \frac{\sqrt{2}M_0}{I_2}\sqrt{\mathbb{E}\left[(\hat{I}_{2,N} - I_2)^2\right]},
$$

$$(16)$$

where we use the fact that $G_1(Z)/G_2(Z) = p_{\mathcal{Z}}(D' \mid Z) \leq M_0$ under Assumption 3. Regarding the RMSE of $\hat{I}_{1,N}$ and $\hat{I}_{2,N}$, He et al. (2023) states that under boundary growth conditions for $G_1$ and $G_2$, the error bound

$$
\sqrt{\mathbb{E}\left[(\hat{I}_{j,N} - I_j)^2\right]} = \mathcal{O}(N^{-1+\epsilon}), \quad j \in \{1, 2\} \tag{17}
$$

holds for any arbitrarily small $\epsilon > 0$. Therefore, the RMSE of the RQMC-IS estimator is of order $\mathcal{O}(N^{-1+\epsilon})$. □

## B  SYNTHETIC DATA EXPERIMENTAL DETAILS

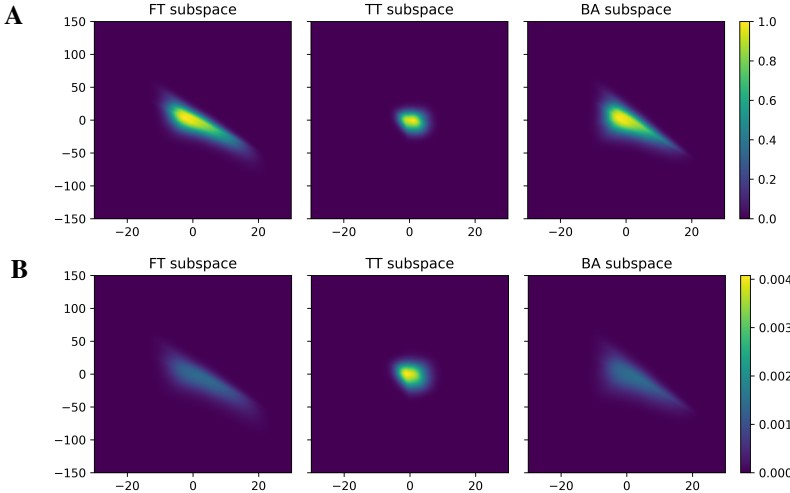

Figure 6: Heat maps across different subspaces with $M = 20$ evaluated on training data. **A.** Likelihood $p_{\mathcal{Z}}(D \mid z)$ **B.** Posterior $p_{\mathcal{Z}}(z \mid D)$. Both the full trajectory subspace and the block-averaging subspace exhibit larger regions of high density in the likelihood and posterior distributions, suggesting a higher quality of weights.

In synthetic examples, we generated synthetic data by computing the scalar output $f(x)$ from a randomly initialized model over 400 points sampled from the intervals $[-7.2, -4.8]$, $[-1.2, 1.2]$ and $[4.8, 7.2]$. Gaussian noise was then added to these outputs to generate the final targets $y$. We followed the setup introduced by Izmailov et al. (2020), using a model with 4 hidden layers of sizes $[200, 50, 50, 50]$ and using the inputs $[x, x^2]$. We used a learning rate of $10^{-2}$, a batch size of 500, and momentum of 0.95. The prior distribution for the weights is set as a Gaussian distribution with a mean of 0 and a standard deviation of 1 for each component. The model is trained for 3000 steps, and the SGD trajectory is collected starting from step 2000. We construct subspaces using a $n = 1000$ point trajectory and memory cost $M = 20$. We employ the tempered posterior given by

$$
p_T(z \mid D) \propto p(D \mid z)^{1/T} p(z) \tag{18}
$$

with a temperature $T = 1.5$.

Figures 1 and 6 present the heat maps of the likelihood and the posterior across different subspaces for testing and training data with $M = 20$. The results demonstrate that the BA subspace closely approximates the FT, and compared to the TT, it contains more 'higher-likelihood' points and a broader posterior region. This indicates that BA aligns more closely with both the training and testing data compared to TT.

## C  UCI REGRESSION EXPERIMENTAL DETAILS

For the UCI small regression experiments, we follow the setup from Wilson et al. (2016), where each dataset is split randomly into 90% training data and 10% testing data, and we employ a fully-connected network with a single hidden layer of 50 units. We perform experiments with 20 different seeds to ensure robustness across random initializations. In each experiment, we use a learning rate of $10^{-3}$, a batch size of 100, momentum of 0.8, a weight decay of $10^{-3}$, and the temperature $T = 10$. The prior distribution for the weights is set as a Gaussian distribution with a mean of 0 and a standard deviation of 100 for each component.

For the UCI-Large datasets, following the set-up of Wilson et al. (2016) and Izmailov et al. (2020), we use a feedforward network with hidden layers of sizes $[10000, 1000, 500, 50, 2]$ and ReLU activations except `skillcraft`, where we employ a smaller architecture with hidden layers of sizes $[1000, 500, 50, 2]$. We set the temperature $T = 1000$ for all datasets except `skillcraft`, where a lower temperature $T = 100$ is used. For the RQMC-IS method, we use Gaussian with a mean of 0 and a standard deviation of 50 as proposal. All models are trained for 1000 steps, and the SGD trajectory is collected starting from step 900. We construct subspaces using a $n = 100$ point trajectory and memory cost $M = 5$.

Table 8 reports the normalized test RMSE (i.e., the root mean square error computed on normalized data) for different subspace inference methods on the UCI-Small datasets. The results indicate that both BA(ESS) and BA(RQMC) achieve lower test RMSE compared to other methods. Table 9 presents the Test Calibration results for the UCI-Small datasets, which evaluate the proportion of true values that fall within the 95% confidence interval of the predicted mean. The subspace constructed using BA produces values closer to 95% compared to TT, given the same posterior sampling method. We also report the normalized test RMSE for the UCI-Large datasets in Table 11, and the Test Calibration results in Table 12.

Table 8: Normalized test RMSE on UCI-Small datasets.

| Dataset | TT (ESS) | BA (ESS) | TT (NUTS) | BA (NUTS) | TT (VI) | BA (VI) | TT (RQMC) | BA (RQMC) |
|---|---|---|---|---|---|---|---|---|
| boston | 0.357±0.081 | **0.355±0.079** | 0.390±0.113 | 0.376±0.091 | **0.356±0.081** | **0.355±0.081** | 0.356±0.080 | **0.356±0.080** |
| concrete | **0.327±0.031** | **0.327±0.030** | 0.342±0.033 | 0.336±0.032 | 0.328±0.030 | 0.329±0.030 | **0.327±0.031** | **0.327±0.030** |
| energy | **0.162±0.030** | **0.162±0.030** | 0.167±0.031 | 0.168±0.030 | 0.164±0.030 | 0.167±0.030 | **0.162±0.030** | **0.162±0.030** |
| naval | 0.075±0.025 | **0.072±0.024** | 11.373±6.478 | 9.784±8.659 | 0.074±0.025 | 0.075±0.031 | 0.076±0.023 | **0.072±0.025** |
| yacht | 0.127±0.033 | 0.126±0.033 | 0.162±0.077 | 0.246±0.278 | 0.125±0.034 | **0.123±0.033** | 0.126±0.033 | **0.124±0.033** |

Table 9: Test calibration on UCI-Small datasets.

| Dataset | TT (ESS) | BA (ESS) | TT (NUTS) | BA (NUTS) | TT (VI) | BA (VI) | TT (RQMC) | BA (RQMC) |
|---|---|---|---|---|---|---|---|---|
| boston | 0.852±0.056 | 0.852±0.057 | **0.967±0.027** | **0.966±0.063** | 0.857±0.060 | 0.867±0.057 | 0.857±0.052 | 0.849±0.054 |
| concrete | 0.917±0.028 | 0.919±0.026 | **0.940±0.030** | 0.926±0.026 | 0.925±0.028 | 0.927±0.033 | 0.917±0.029 | 0.916±0.030 |
| energy | **0.952±0.022** | **0.953±0.019** | 0.960±0.020 | 0.960±0.020 | **0.951±0.022** | **0.953±0.023** | 0.952±0.021 | 0.953±0.019 |
| naval | 0.983±0.008 | **0.980±0.006** | 0.664±0.215 | 0.761±0.213 | **0.980±0.007** | **0.979±0.006** | 0.983±0.006 | 0.982±0.006 |
| yacht | 0.976±0.025 | **0.966±0.030** | 0.997±0.010 | 0.997±0.010 | 0.974±0.026 | 0.976±0.027 | 0.976±0.025 | **0.968±0.025** |

## D  IMAGE CLASSIFICATION EXPERIMENTAL DETAILS

We evaluate our methods on the `CIFAR` datasets. Following Izmailov et al. (2020), we conduct experiments using both VGG-16 and PreResNet164 architectures, trained on `CIFAR10` and `CIFAR100`. For VGG-16, we use a learning rate of $5 \times 10^{-2}$, momentum of 0.9, and weight decay of $5 \times 10^{-4}$.

Table 10: Computational cost on UCI-Small datasets. The cost is measured by the number of forward passes through the model on the training set.

| Dataset | TT (ESS) | BA (ESS) | TT (NUTS) | BA (NUTS) | TT (VI) | BA (VI) | TT (RQMC) | BA (RQMC) |
|---|---|---|---|---|---|---|---|---|
| boston | 2427.1±146.2 | 1951.8±87.0 | 1815.7±242.3 | 1860.1±202.2 | 2000 | 2000 | **1024** | **1024** |
| concrete | 1931.4±154.0 | 1617.3±124.5 | 13295.0±84.6 | 13293.8±109.3 | 2000 | 2000 | **1024** | **1024** |
| energy | 2432.1±128.8 | 1912.4±95.9 | 1532.9±153.7 | 2060.4±162.7 | 2000 | 2000 | **1024** | **1024** |
| naval | 2824.9±244.8 | 2140.3±149.7 | 10831.7±2110.5 | 8964.0±3222.4 | 2000 | 2000 | **1024** | **1024** |
| yacht | 2574.4±110.7 | 2258.5±103.0 | 2071.9±209.8 | 2096.6±164.0 | 2000 | 2000 | **1024** | **1024** |

Table 11: Normalized test RMSE on UCI-Large datasets.

| Dataset | TT (ESS) | BA (ESS) | TT (NUTS) | BA (NUTS) | TT (VI) | BA (VI) | TT (RQMC) | BA (RQMC) |
|---|---|---|---|---|---|---|---|---|
| elevators | **0.349±0.008** | **0.347±0.007** | 0.783±0.292 | 0.745±0.166 | **0.347±0.008** | **0.347±0.008** | **0.348±0.007** | **0.348±0.007** |
| protein | 0.633±0.010 | **0.629±0.009** | 1.057±0.081 | 1.019±0.124 | 0.635±0.009 | 0.637±0.008 | 0.632±0.009 | **0.629±0.009** |
| pol | **0.399±0.021** | **0.399±0.021** | 1.507±0.407 | 1.460±0.916 | 0.436±0.024 | 0.441±0.026 | **0.400±0.021** | **0.399±0.021** |
| keggD | **0.095±0.033** | **0.095±0.031** | 0.102±0.033 | 0.100±0.028 | **0.095±0.030** | **0.095±0.030** | **0.095±0.032** | **0.095±0.031** |
| keggU | 0.124±0.004 | 0.124±0.004 | 0.126±0.004 | 0.126±0.005 | **0.122±0.003** | **0.122±0.003** | 0.125±0.004 | **0.123±0.003** |
| skillcraft | 0.650±0.030 | 0.648±0.028 | 0.930±0.266 | 0.844±0.157 | **0.647±0.027** | **0.647±0.027** | 0.646±0.027 | 0.646±0.027 |

Table 12: Test calibration on UCI-Large datasets.

| Dataset | TT (ESS) | BA (ESS) | TT (NUTS) | BA (NUTS) | TT (VI) | BA (VI) | TT (RQMC) | BA (RQMC) |
|---|---|---|---|---|---|---|---|---|
| elevators | 0.937±0.007 | 0.934±0.008 | **0.952±0.027** | 0.938±0.031 | 0.933±0.007 | 0.933±0.007 | 0.942±0.007 | 0.938±0.007 |
| protein | **0.946±0.005** | 0.945±0.004 | 0.992±0.006 | 0.989±0.011 | 0.943±0.004 | 0.943±0.004 | **0.946±0.004** | **0.946±0.004** |
| pol | **0.988±0.004** | 0.989±0.004 | 1.000±0.000 | 0.997±0.013 | 0.992±0.002 | 0.993±0.002 | **0.988±0.004** | **0.988±0.004** |
| keggD | **0.962±0.003** | 0.962±0.002 | 0.974±0.007 | 0.974±0.006 | 0.964±0.003 | 0.965±0.002 | **0.962±0.003** | **0.962±0.002** |
| keggU | 0.969±0.005 | 0.968±0.006 | 0.969±0.006 | 0.968±0.004 | **0.965±0.003** | **0.965±0.002** | 0.971±0.007 | 0.968±0.006 |
| skillcraft | 0.958±0.010 | 0.957±0.013 | 0.973±0.020 | 0.975±0.021 | 0.945±0.014 | 0.945±0.014 | 0.953±0.011 | **0.951±0.011** |

Table 13: Computational cost on UCI-Large datasets. The cost is measured by the number of forward passes through the model on the training set.

| Dataset | TT (ESS) | BA (ESS) | TT (NUTS) | BA (NUTS) | TT (VI) | BA (VI) | TT (RQMC) | BA (RQMC) |
|---|---|---|---|---|---|---|---|---|
| elevators | 1206.9±153.1 | **811.2±27.8** | 1981.2±252.0 | 2173.6±256.8 | 2000 | 2000 | 1024 | 1024 |
| protein | 2005.2±235.3 | 1238.7±91.7 | 2430.2±266.7 | 2226.5±198.2 | 2000 | 2000 | **1024** | **1024** |
| pol | 1227.9±49.7 | 1059.8±41.4 | 2582.0±485.6 | 2214.5±296.3 | 2000 | 2000 | **1024** | **1024** |
| keggD | 1945.1±188.3 | 1354.1±85.2 | 1809.1±381.3 | 1856.7±143.5 | 2000 | 2000 | **1024** | **1024** |
| keggU | 1952.2±450.7 | 1503.8±446.7 | 1825.9±305.7 | 2024.9±404.2 | 2000 | 2000 | **1024** | **1024** |
| skillcraft | 1845.1±196.5 | 1499.3±127.3 | 2068.2±453.7 | 1815.9±313.8 | 2000 | 2000 | **1024** | **1024** |

Table 14: Bayes factors and testing data evidence ratios on CIFAR datasets (Tail trajectory subspace against Block-averaging subspace).

| Dataset | VGG-16 on CIFAR10 | PreResNet164 on CIFAR10 | VGG-16 on CIFAR100 | PreResNet164 on CIFAR100 |
|---|---|---|---|---|
| Bayes factor | 0.280 ± 0.031 | 0.270 ± 0.054 | 0.227 ± 0.004 | 0.381 ± 0.026 |
| Evidence ratios | 0.193 ± 0.031 | 0.285 ± 0.122 | 0.111 ± 0.016 | 0.353 ± 0.040 |

Table 15: Negative log likelihood (NLL) on CIFAR datasets.

| Models | TT (ESS) | BA (ESS) | TT (VI) | BA (VI) | BA (RQMC) |
|---|---|---|---|---|---|
| VGG-16 on CIFAR10 | 0.304±0.015 | 0.307±0.012 | 0.301±0.011 | 0.346±0.022 | 0.306±0.015 |
| PreResNet164 on CIFAR10 | 0.188±0.009 | 0.193±0.009 | 0.189±0.002 | 0.214±0.011 | 0.192±0.008 |
| VGG-16 on CIFAR100 | 1.876±0.038 | 1.850±0.080 | 1.674±0.066 | 2.018±0.038 | 1.885±0.039 |
| PreResNet164 on CIFAR100 | 0.936±0.017 | 0.976±0.011 | 0.881±0.008 | 0.964±0.009 | 0.901±0.003 |

Table 16: Classification accuracy (ACC(%)) on corrupted CIFAR datasets using VGG-16.

| Severity | TT (ESS) | BA (ESS) | TT (VI) | BA (VI) | BA (RQMC) |
|---|---|---|---|---|---|
| 1 | **91.30±0.23** | **91.32±0.20** | 91.09±0.25 | **91.28±0.28** | **91.32±0.26** |
| 2 | **89.42±0.29** | 89.34±0.25 | 89.19±0.41 | 89.33±0.31 | **89.39±0.29** |
| 3 | 86.18±0.40 | 86.15±0.42 | 85.87±0.49 | **86.26±0.48** | 86.21±0.33 |
| 4 | **81.04±0.77** | 80.96±0.82 | 80.54±0.86 | **81.04±0.88** | 80.94±0.70 |
| 5 | 64.78±1.57 | 64.54±1.79 | 64.12±1.76 | **64.93±1.79** | 64.62±1.58 |

Table 17: AUC(%) for OOD detection on CIFAR-SVHN.

| Models | TT (ESS) | BA (ESS) | TT (VI) | BA (VI) | BA (RQMC) |
|---|---|---|---|---|---|
| VGG-16 on CIFAR10 | 87.91±2.35 | 88.86±1.17 | 85.93±1.77 | **89.86±1.25** | 88.06±1.53 |
| PreResNet164 on CIFAR10 | 94.04±1.06 | **94.26±0.40** | 93.28±1.80 | 94.04±0.69 | 93.67±0.26 |
| VGG-16 on CIFAR100 | 78.85±0.86 | 78.75±1.26 | **78.90±0.79** | 78.23±1.22 | **78.87±1.38** |
| PreResNet164 on CIFAR100 | **77.43±2.30** | 77.07±1.71 | 76.59±2.72 | 77.03±1.74 | 76.90±2.05 |

For PreResNet164, we use a learning rate of $10^{-1}$ and weight decay of $3 \times 10^{-4}$. We set the batch size to 500 and randomly split 10% of the data for testing. All models are trained for 300 steps, with the SGD trajectory collected starting from step 160. We construct subspaces using a 140-point trajectory with a memory cost of $M = 5$ and set the temperature $T = 1000$ for predictive inference. All experiments are repeated five times for robustness.

Table 14 reports the Bayes factors and evidence ratios for different methods, showing substantial evidence in favor of the BA subspace over the TT subspace. For the predictive performance, we report the classification accuracy in Table 6 and the corresponding negative log-likelihood in Table 15. For noisy data, we present the classification accuracy of PreResNet164 and VGG-16 across different severity levels in Tables 7 and 16, respectively. For OOD detection, we report the AUC scores on the SVHN dataset in Table 17.

# E  STATEMENT ON COMPUTING RESOURCES

Our numerical experiments were conducted on a Linux server equipped with six nvidia RTX A6000 graphics cards and a pair of 20-core Intel 5218R CPUs.

