# OpenReview forum: "Evaluating and Improving Subspace Inference in Bayesian Deep Learning"
_ICLR.cc/2025/Conference — Submitted to ICLR 2025_

### Official Review · Reviewer_K1aU · 2024-11-01

**Soundness:** 3
**Presentation:** 3
**Contribution:** 2
**Rating:** 5
**Confidence:** 3

**Summary:**

This paper proposed 3 modifications to the existing subspace Bayesian inference methods, targeting at (1) subspace construction; (2) direct subspace quality evaluation; and (3) subspace sampling. Specifically, the author improves the tail trajectory subspace construction (only take the last few samples) by taking spread samples across the entire trajectory, covering broader range of dynamics. The author also propose direct evaluation of subspace quality based on the Bayes factor (evidence likelihood ratio). Last but not least, the author also proposed to use important sampling or quasi-MC for subspace sampling. Empirically, the proposed method is tested on UCI and image classification, demonstrating the improved performance compared to the existing one.

**Strengths:**

The paper is clearly written and very easy to follow. The idea of using separated samples to reconstruct the subspace is straight-forward but effective. The combination of block-averaging (BA) subspace reconstruction and quasi-MC is simple but seems to be effective empirically.

**Weaknesses:**

The overall methodology seems to be a straightforward combination of three existing (with minor modifications) approaches. To demonstrates its benefits, it would be great to cite/perform comparisons with other Bayesian inference techniques.

Another questions I have is during the subspace reconstruction, do you need to flatten the matrix from $mxn$ to $d$? Doesn't this destroy the structural information stored in the original weight matrix? For example, matrix multiplication Wx represents each row of W is dot product with x, this is a kind of structural information stored within the matrix. If you flatten this, you loose such info. Is this because flatten W and then perform SVD can give you a much lower low-dimensional $z$, compared to performing SVD on the matrix W?

What if you keep the matrix structure, but perform SVD on the matrix trajectory to get USV^T, and treat S as your Z?

For the image classification and UCI, I am curious about the full trajectory performance. Why don't you report it?

**Questions:**

See above.

---

> ### Author Response · Authors · 2024-11-24
> **Response**
>
> Dear reviewer K1aU, we sincerely thank you for your detailed advice and the time you spent on our paper. Here are our replies to your comment:
>
> ---
>
> **Weakness:**
>
> **W1:** The overall methodology seems to be a straightforward combination of three existing (with minor modifications) approaches.
>
> **A1:** Thank you for your comment. Our work introduces (1) a novel subspace construction method (block-averaging) that captures global trajectory efficiently, (2) new metrics (Bayes factors and evidence ratios) to evaluate subspace quality—an overlooked aspect in prior work—and (3) an efficient approach for posterior predictive computation in low-dimensional subspaces. These contributions go beyond simply combining existing methods and address limitations in prior approaches.
>
> **W2:** During the subspace reconstruction, do you need to flatten the matrix from $mxn$ to $d$? What if you keep the matrix structure, but perform SVD on the matrix trajectory to get $USV^\top$, and treat $S$ as your $Z$?
>
> **A2:** Thank you for your question. For all model weights, we flatten them into a $d$-dimensional vector like prior works [1, 2]. While we acknowledge that this may lose some structural information, retaining the matrix structure (e.g., performing layer-wise SVD) introduces challenges. For instance, different layers have weight matrices of varying sizes, making it unclear how to combine these layers within a unified subspace. For example, VGG-16 has about 138 million parameters across layers of varying dimensions, which complicates maintaining matrix structures for subspace construction.
> If there are relevant studies addressing these issues, we would be glad to explore them further.
>
> **W3:** For the image classification and UCI, I am curious about the full trajectory performance. Why don't you report it?
>
> **A3:** Performing PCA or randomized SVD on the full trajectory requires storing all $n$ trajectories. When the dimensionality of model weights is high, the memory and computational costs become prohibitively expensive, making full trajectory subspace construction infeasible in such scenarios. We have included this clarification in the revised manuscript (Lines 496-497). Thank you for your feedback.
>
> ---
>
> [1]. Li et al. "Measuring the intrinsic dimension of objective landscapes", 2018.
>
> [2]. Izmailov et al. "Subspace inference for Bayesian deep learning", 2020.

---

> ### Comment · Reviewer_K1aU · 2024-12-02
>
> Thanks for the author's feedback, the response address some of my concerns. However, the block average is one simple extension (i.e. take samples with certain interval) that is not difficult to think of; the new metrics Bayes' factors and evidence ratios are also based on marginal likelihood, which has been used for model selection before. Therefore, these contribution to methodology seems to be a bit incremental. Therefore, I keep the original score.

---

### Official Review · Reviewer_hUx8 · 2024-11-02

**Soundness:** 3
**Presentation:** 3
**Contribution:** 3
**Rating:** 6
**Confidence:** 4

**Summary:**

This work proposes a bayesian subspace inference method, focusing on three aspects, i.e., subspace construction, subspace evaluation, and inference efficiency. Correspondingly, the block-averaging scheme, bayes factor construction, and some importance sampling techniques are leveraged. In general, the problem studied in this work is interesting, however, the current content does not show significantly appealing advantages. Thus, the following comments in the review system are raised. I would be willing to raise my scores if my concerns or possible misunderstandings are well addressed in the rebuttal.

**Strengths:**

1. The problems addressed are interesting.

2. The paper is easy to read and the proposed method is quite relevant to many researches in  DL optimization.

3. The proposed evaluation metric appears to be the most interesting aspect to me, personally.

**Weaknesses:**

Weakness
1. The BA scheme seems not sufficiently novel, as I saw it somewhat like downsampling to the full trajectory, which is closely related to some existing works (see questions below).

2. Despite the importance and efficacy of the proposed evaluation metric, it feels hard to find the practical use and values in helping the optimization or inferencee process, rather than simply being a metric that shows one method outperforms one another, because we can also just compare the inference results with other metrics to compare  (see questions below).

3. In the current numerical experiments, the evidence to the inference efficiency is insufficient in comprehensive evaluations, as it is claimed as one of the main contribution of this work  (see questions below).

**Questions:**

1. Is it appropriate to interpret the proposed subspace construction method as a kind of downsampling scheme to the FT method?

There are some recent literature missing in this work, e.g., SWA [1], TWA [2], DLDR [3]. In [1], the algorithm can be regarded as special case for TT. In [2], the subspace can be flexibly constructed by utilizing the checkpoints in the training head stage, tail stage, and even the fine tuning stage, the later of which meaning that a downsampling scheme can be applied to FT, to some extend. In [3], it samples some checkpoints in the trajectory, commonly in the stage where a descent performance is already gained (informally saying a downsampling to the “almost FT”).

Please provide more explanations on the novelty and propertied of the proposed methods, and also more discussions w.r.t. the existing work, i.e.,

- explain how the block-averaging (BA) method differs from or improves upon SWA, TWA, and DLDR.

- provide a more thorough comparison table or discussion that highlights the key differences and potential advantages of BA over these existing methods.

-  clarify the novel aspects of BA that go beyond simple downsampling, if any.

 [1] P. Izmailov, D. Podoprikhin, T. Garipov, D. Vetrov, and A. G. Wilson, Averaging weights leads to wider optima and better generalization, UAI, 2018.

[2] T. Li, Z. Huang, Q. Tao, Y. Wu, and X. Huang, Trainable weight averaging: Efficient training by optimizing historical solutions, ICLR, 2023.

[3] T. Li, L. Tan, Z. Huang, Q. Tao, Y. Liu, and X. Huang, Low dimensional trajectory hypothesis is true: DNNs can be trained in tiny subspaces, IEEE TPAMI, 2022.



2. If the answer to question 1 is yes. The results in table 1 is obvious, so I don’t see much importance of taking much length showing table 1.

Nonetheless, it is really interesting to see the results in Figure 3, showing the significance of the proposed evaluation metric on the condition that the testing data evidence ratios are informative comparison baselines.  Since it is proposed as an evaluation metric on the quality of the constructed subspace. However, during the construction or some tuning towards obtaining the subspace, I didn’t notice how such metric is utilized, e.g., we can leverage such metric is determine the dimensions $k$ or $M$, etc. From the existing results, I only saw that this metric is simply used to show that BA is better than TT, which is less informative, as anyways we can use inference performance metrics to compare.

Please elaborate the utility of the proposed metric, i.e.,
- demonstrate how their proposed metric (Bayes factor and evidence ratio) can be used to guide hyperparameter selection, such as choosing optimal values for k or M.

- discuss potential applications of these metrics beyond just comparing BA to TT, such as in model selection or uncertainty quantification tasks or possibly provide examples of using these metrics during the subspace construction process to iteratively improve subspace quality.

- explain how these metrics offer insights that traditional performance metrics may not capture.


3. Some minor aspects: the computational cost of obtaining this metric can also be explained with some details; In table 2, I did’t see much advantage of BA over TT and FT; rather than comparing the performance and efficiency jointly in a simple table, I would suggest to have some comparisons specifically on the efficiency.

- a more detailed analysis of the computational costs associated with calculating the proposed evaluation metrics.

- some separate demonstration that focuses solely on comparing the computational efficiency of BA, TT, and FT methods.

- discuss potential reasons for the non-distinctively advantageous performance of BA in Table 2; are there any trade-offs involved?

---

> ### Author Response · Authors · 2024-11-24
> **Response**
>
> Dear reviewer hUx8, we sincerely thank you for your detailed advice and the time you spent on our paper. Here are our replies to your comment:
>
> ---
>
> **Questions:**
>
> **Q1:** Is it appropriate to interpret the proposed subspace construction method as a kind of downsampling scheme to the FT method?
>
> **A1:**  Thank you for raising this insightful point. We agree that BA can be viewed as a form of downsampling applied to the full trajectory method. We now describe this interpretation in the revised manuscript as follows (Line 183):
>
> > BA can be viewed as a structured downsampling of the full trajectory, providing a similar subspace while significantly reducing computational and memory costs, as shown in Figure 1.
>
> **Q2:** How the block-averaging (BA) method differs from or improves upon SWA, TWA, and DLDR?
>
> **A2:**  SWA, TWA, and DLDR focus on finding a **single** weight $\hat{w}$ based on the SGD trajectory. For instance, SWA averages weights to achieve better generalization, while TWA identifies optimal weights within the linear space spanned by $n$ trajectories. DLDR leverages a quasi-Newton-based algorithm to find an optimal solution within the subspace.
>
> In contrast, BA (along with TT and FT) aims to find a **set** of weights residing in some subspace $\{w: \hat{w} + P z, z\in \mathbb{R}^k\}$. Within such a subspace, inference algorithms (like MCMC, variational inference) are computationally more efficient, and predictions (using all points in the subspace) are combined through **Bayesian model averaging**. In this work, we propose BA, which constructs a high-quality subspace with limited computational cost, along with some principled subspace quality evaluation methods. We appreciate the reviewer highlighting these related works like SWA, TWA and DLDR, and we have included discussions in the Related Works paragraph (Lines 74-76).
>
> **Q3:** Clarify the novel aspects of BA that go beyond simple downsampling.
>
> **A3:** The novelty of BA lies in: (1) approximating FT while significantly reducing memory and computational costs; (2) building on SWA’s observation in [1] that averaging weights can improve generalization, BA partitions the trajectory into equidistant blocks and computes block averages, effectively preserving global variability; and (3) ensuring the resulting subspace aligns with PCA interpretations through SVD. Additionally, our work also introduces methods to evaluate subspace quality and efficiently perform predictive computations.
>
> **Q4:** The results in Table 1 is obvious, so I don’t see much importance of taking much length showing Table 1.
>
> **A4:** Thank you for the suggestion. We will adjust the discussion of Table 1 accordingly. The purpose of Table 1 is to demonstrate that BA can achieve results close to FT with lower cost, while TT shows significant angular distance in the first principal components compared to both FT and BA.
>
> **Q5:** How the proposed metric (Bayes factor and evidence ratio) can be used to guide hyperparameter selection?
>
> **A5:** Thank you for your suggestion. We agree that the Bayes factor and evidence ratio may provide some guidance in hyperparameter selection. For example, Figure 3 indicates that for the BA subspace, $M=5$ is a good choice as it achieves the highest Bayes factor and evidence ratio. While our current work focuses on demonstrating the utility of these metrics for evaluating subspace quality (see **A6** for details), applying them for hyperparameter tuning is an important direction in future work.
>
> **Q6:** Please elaborate the utility of the proposed metrics, and explain how these metrics offer insights that traditional performance metrics may not capture.
>
> **A6:** Bayes Factors and evidence ratios can apply to any subspaces or subsets of original weights space, not just BA and TT. Bayes Factors evaluates subspaces using only training data, which is often overlooked in current practices for assessing fit without test data. Evidence ratios like prior predictive metrics, do not rely on the posterior derived from training data. Both metrics are computationally cheaper (see **A8** for details) than the predictive performance metrics mentioned by the reviewer.

---

> > ### Author Response · Authors · 2024-11-24
> > **Response Contd.**
> >
> > **Q7:** In table 2, I did’t see much advantage of BA over TT and FT. Discuss potential reasons for the non-distinctively advantageous performance of BA in Table 2.
> >
> > **A7:** Thank you for the comment. The true posterior predictive $p_\mathcal{Z}(D' \mid D)$ differs across subspaces $\mathcal{Z}$. The purpose of Table 2 is to compare the accuracy and computational cost of RQMC-IS against other sampling methods within each subspace, rather than to evaluate the differences between subspaces. The comparison of subspaces in this task is provided in Figures 3 and 4.
> >
> > **Q8:** I would suggest to have some comparisons specifically on the computational efficiency of BA, TT, and FT methods, and computational costs associated with calculating the proposed evaluation metrics.
> >
> > **A8:**  Thank you for the suggestion. The computational and memory costs for subspace construction in BA ($\mathcal{O}(Md)$ memory and $\mathcal{O}(Md \log K + (M + d) K^2)$ computation) are the same as TT.
> >
> > For our proposed evaluation metrics, Bayes factors (Eq. (6)) and evidence ratios (Eq. (7)) rely solely on likelihood evaluations on the training and testing sets, respectively.
> > In contrast, posterior prediction (Eq. (5)) not only requires sampling from the posterior on the training set (which is often significantly more expensive than simple likelihood evaluations), but also requires likelihood evaluations on the test data. This makes posterior predictive more expensive than Bayes factors and evidence ratios.
> >
> > ---
> >
> > [1]. Izmailov et al. "Averaging weights leads to wider optima and better generalization", 2018.

---

> > ### Comment · Reviewer_hUx8 · 2024-11-26
> > **Thanks for the response.**
> >
> > I would like to thank the authors for the responses in the rebuttal.
> >
> > Most of my concerns have been responded, and few are remained.
> > - For A2, it is till a bit unclear, as what I refer is the construction of the projection matrix $P$ leading to the subspace that it projects to, so all these methods aim to find some weights residing in such a constructed space. Other mentioned works also have such projection matrix for the subspace, and the only difference is that how they compute such $P$. In general $P$ is computed from the trajectory checkpoints, so it differs in the choices of those checkpoints and their processing, meaning that anyways some $P$ is given by SVD on some $W$. From Algorithm 1, it seems the same in this spirit. Nonetheless the proposed methods is under Bayesian frameworks, while [1-3] are deterministic.
> >
> > - Since the utilization and evaluation of the proposed Bayes factor and evidence ratio are similar to the submitted version. For the aspect, I may remain my evaluation, but I would  look forward to future work.
> >
> > Below is some further advice.
> > - Currently, only algorithm 1 is provided for subspace construction steps. It would be nice to have a complete steps of both algorithm 1 and the Bayes part in a comprehensive algorithm box with details.
> > -  It would nice to  release of the codes in the supplements upon possibilities and regulations on github, where details in explanations and comparisons w.r.t. other bayesian methods and other subspace constructions can be provided for practitioners

---

> > > ### Author Response · Authors · 2024-11-28
> > > **Thanks for the feedback**
> > >
> > > Here are our replies to your comment:
> > >
> > > ---
> > >
> > > **Comments**:
> > >
> > > **C1:** From Algorithm 1, it seems the same in this spirit. Nonetheless the proposed methods is under Bayesian frameworks, while [1-3] are deterministic.
> > >
> > > **A1:** Thank you for the comment. We do not consider SWA [1] to construct a subspace, as it averages different weights together without explicitly defining a subspace.
> > >
> > > We agree that DLDR [3], TT, and BA share similarities in subspace construction: all derive a matrix from the trajectory through a form of downsampling, followed by SVD to obtain the projection matrix $P$. The key difference lies in the downsampling strategy, i.e., TT uses equidistant points from the trajectory tail, DLDR selects random points, and BA computes block averages across the entire trajectory. However, as shown in Table 1 and Figure 1, different downsampling strategies can result in substantially different subspaces, which motivated us to explore how model evidence can be used to evaluate the quality of these subspaces.
> > >
> > > **C2:** Currently, only algorithm 1 is provided for subspace construction steps. It would be nice to have a complete steps of both algorithm 1 and the Bayes part in a comprehensive algorithm box with details.
> > >
> > > **A2:** Thank you for your suggestion. We have added the complete steps in a comprehensive algorithm in the updated Appendix A.
> > >
> > > ---
> > >
> > > [1]. Izmailov et al. "Averaging weights leads to wider optima and better generalization", 2018.
> > >
> > > [2]. Li et al. "Trainable weight averaging: Efficient training by optimizing historical solutions", 2023
> > >
> > > [3]. Li et al. "Low dimensional trajectory hypothesis is true: DNNs can be trained in tiny subspaces", 2022

---

> > > > ### Author Response · Authors · 2024-11-28
> > > > **Response Contd.**
> > > >
> > > > We provide the full algorithm that will be added to Appendix A in the revised manuscript.
> > > >
> > > > ---
> > > >
> > > > ## Appendix A - Complete Algorithm: Subspace Construction and Uncertainty Quantification
> > > >
> > > > Algorithm 2 presents a complete algorithm that integrates our proposed subspace inference method. The algorithm is divided into three key steps: Step 1 focuses on subspace construction, where the trajectory obtained from training is used to construct a low-dimensional subspace. Step 2 introduces subspace evaluation, leveraging model evidence metrics to assess subspace quality, which is an important step that is often overlooked in current practices. Step 3 describes inference methods, where RQMC-IS is employed for efficient predictive estimation within the low-dimensional subspace.
> > > >
> > > > ---
> > > >
> > > > **Algorithm 2:** Subspace Construction and Uncertainty Quantification
> > > >
> > > > ---
> > > >
> > > > **Input:** training dataset $D$, testing dataset $D^\prime$, prior distribution for weights $p(w)$, proposal distribution on subspace $q(z)$, number of sampled weights $N$;
> > > >
> > > > **Step 1: Subspace Construction**
> > > >
> > > > ​ ​ ​ ​ Get subspace $\mathcal{Z} = \{ \hat{w} + P z \mid z \in \mathbb{R}^k \} $ using Algorithm 1
> > > >
> > > > ​ ​ ​ ​ Derive induced likelihood $p_{\mathcal{Z}}(D \mid z) = p_{\mathcal{W}}(D \mid \hat{w} + Pz)$ and induced prior $p_{\mathcal{Z}}(z) \propto p(\hat{w} + Pz)$
> > > >
> > > > **Step 2: Subspace Evaluation with Marginal Likelihood**
> > > >
> > > > ​ ​ ​ ​ Generate low-discrepancy sequence $\{U_1, \cdots, U_N\}$ from $[0, 1)^k$
> > > >
> > > > ​ ​ ​ ​ **for** $i \in [1, 2, \cdots, N]$ **do**
> > > >
> > > > ​ ​ ​ ​ ​ ​ ​ ​ Transform samples using inverse CDF: $z_i \leftarrow F_q^{-1}(U_i)$
> > > >
> > > > ​ ​ ​ ​ ​ ​ ​ ​ Compute importance weights: $w_i \leftarrow {p_\mathcal{Z}(z_i)} / {q(z_i)}$
> > > >
> > > > ​ ​ ​ ​ **end for**
> > > >
> > > > ​ ​ ​ ​ Estimate marginal likelihood on $D$ and $D^\prime$:
> > > >
> > > > ​ ​ ​ ​ ​ ​ ​ ​ $\hat p_{\mathcal{Z}}(D) \leftarrow \frac{\sum_{i=1}^N w_i p_\mathcal{Z}(D \mid z_i)}{\sum_{i=1}^N w_i},\quad\hat p_{\mathcal{Z}}(D^\prime) \leftarrow \frac{\sum_{i=1}^N w_i p_\mathcal{Z}(D^\prime \mid z_i)}{\sum_{i=1}^N w_i}$
> > > >
> > > > **Step 3: Uncertainty Quantification using RQMC-IS**
> > > >
> > > > ​ ​ ​ ​ Generate low-discrepancy sequence $\{\tilde U_1, \cdots, \tilde U_N\}$ from $[0, 1)^k$
> > > >
> > > > ​ ​ ​ ​ **for** $i \in [1, 2, \cdots, N]$ **do**
> > > >
> > > > ​ ​ ​ ​ ​ ​ ​ ​ Transform samples using inverse CDF: $\tilde z_i \leftarrow F_q^{-1}(\tilde U_i)$
> > > >
> > > > ​ ​ ​ ​ ​ ​ ​ ​ Compute importance weights: $\tilde w_i \leftarrow p_\mathcal{Z}(D \mid \tilde z_i) p_\mathcal{Z}(\tilde z_i) / q(\tilde z_i)$
> > > >
> > > > ​ ​ ​ ​ **end for**
> > > >
> > > > ​ ​ ​ ​ Estimate RQMC-IS based posterior predictive:
> > > >
> > > > ​ ​ ​ ​ ​ ​ ​ ​ $\hat p_{\mathrm{RQMC}}(N,q; D,D^{\prime}) \leftarrow \frac{\sum_{i=1}^N \tilde w_i p_\mathcal{Z}(D^\prime \mid \tilde z_i)}{\sum_{i=1}^N \tilde w_i}$
> > > >
> > > > **Output:** Marginal likelihood estimator $\hat p_{\mathcal{Z}}(D)$ and $\hat p_{\mathcal{Z}}(D^\prime)$; Posterior predictive estimator $\widehat{p}_{\mathrm{RQMC}}(N,q; D,D^{\prime})$

---

### Official Review · Reviewer_bE76 · 2024-11-03

**Soundness:** 2
**Presentation:** 2
**Contribution:** 2
**Rating:** 3
**Confidence:** 4

**Summary:**

The authors proposed a novel method of constructing low-dimensional subspace (BA) that allows the aggregate entire weights trajectory during SGD rather than the trajectory tail. The authors argue that their method is not only as computationally effective as the Tail Trajectory (TT) method but improves inference quality by better capturing the subspace landscape. Then the authors provide a simple method to evaluate the quality of constructed subspaces based on the Bayes Factor. They apply the proposed estimator and show that the subspaces constructed from the BA trajectory outperform those from TT. At last, the authors propose a new method of Bayesian Inference in the newly constructed subspace. They combine Importance Sampling with the randomized quasi-Monte Carlo method to get an estimator with a better convergence rate.

The authors provide multiple experiments to assess the quality of the proposed subspace as well as the RQMC-IS method and argue that those algorithms achieve higher test accuracy

**Strengths:**

1. The authors provide a full explanation of technical detail, including all architecture details and training process.

**Weaknesses:**

1. The novelty of the proposed algorithm is arguable. The only difference between the proposed Algorithm 1 and Algorithm 2 from [1] is that the latter uses the last point in each block (that is defined by the appropriate choice of hyperparameter c: moment update frequency) when Algorithm 1 uses the mean point in the block.
2. Figure 2 seems to be misleading because both BA and TT subspaces are constructed from the SGD points after some initial warm-up process that is the same in the proposed paper and in [1]. For example, for Synthetic Regression, both methods use the last 1000 points, while for the CIFAR dataset, they use the last 140 points.
3. The conclusion that the proposed BA and RQMC-IS have a better quality is based mostly on Tables 4, 5, 7, 9, 10, 12, 13, 16, and 17, where most of the numbers in each line are within the standard deviation of one another. From my point of view, the provided results don't show any statistically significant difference between BA and TT or between RQMC-IS and ESS/VI. I advise authors to provide a more detailed analysis of the metrics and show that there is a definitive difference between methods. For example, one can provide a similar Figure as Figure 4 from [1].
4. Some of the tables provide incomplete comparisons between methods. For example, TT (RQMC) is missing from Tables 7 and 8. Also, there is no comparison between RQMC and SNIS.
5. There is some inconsistency between the results from the paper and the prior work. For example, Figure 4 is used to argue that compared to the TT subspace, the BA subspace reflects higher uncertainty in data-sparse regions and higher confidence in data-rich regions. However, Figure 4 (middle) should be the same as Figure 3 (ESS, PCA Subspace) from [1], where TT captures uncertainty the same way BA does. Is there any difference in the experiment setup that caused this difference?
6. The proposed sampler RQMC-IS seems to require evaluation of $p(D|z)$ using all training data and couldn't be estimated using mini-batches, which makes this method practically useless for large neural networks and large datasets. At the same time VI can be performed by Doubly Stochastic VI using mini-batches that drastically improve speed. What type of VI did the authors use in their paper? Have they considered comparison with SVI [2] or other scalable methods?

Minor typos:
Lines 207, 242: subapce -> subspace

[1] Pavel Izmailov, Wesley J Maddox, Polina Kirichenko, Timur Garipov, Dmitry Vetrov, and Andrew Gordon Wilson. Subspace inference for bayesian deep learning. In Uncertainty in Artificial Intelligence, pp. 1169–1179. PMLR, 2020.
[2] Hoffman, M. D., Blei, D. M., Wang, C., and Paisley, J. (2013). Stochastic variational inference. The Journal of Machine Learning Research, 14(1):1303–1347.

**Questions:**

Please see the weaknesses for questions and improvements.

---

> ### Author Response · Authors · 2024-11-24
> **Response**
>
> Dear reviewer bE76, we sincerely thank you for your detailed advice and the time you spent on our paper. Here are our replies to your comment:
>
> ---
>
> **Weakness:**
>
> **W1:** The novelty of the proposed algorithm is arguable, both BA and TT subspaces are constructed from the SGD points after some initial warm-up process.
>
> **A1:** While BA and TT both use the SGD points (like SWA), BA introduces a new strategy for finding the subspace direction. The TT strategy in [1] practically used the last $M$ points of the SGD trajectory, because the spacing parameter $c$ is set to 1 in all experiments.
> Even with $c>1$, TT still ignores the majority of the points from the SGD trajectory, and the center of these $M$ points is not $\hat{w}$, meaning the corresponding SVD decomposition no longer aligns with the interpretations of PCA.
> In contrast, BA divides the entire trajectory (after warm-up) into $M$ equidistant blocks and computes the block averages, capturing the global variability over the full trajectory while maintaining the same computational cost as TT.
>
> **W2:** Figure 2 seems to be misleading because both BA and TT subspaces are constructed from the SGD points after some initial warm-up process that is the same in the proposed paper and in [1]. For example, for Synthetic Regression, both methods use the last 1000 points.
>
> **A2:** In Figure 2, we did not include the warm-up trajectory; instead, we drew the trajectories after the warm-up phase for the online update of the mean $\hat{w}$. We believe Figure 2 precisely demonstrates the difference between FT, TT, and BA. For example, for the Synthetic experiment with $M=20$, all methods use the same trajectory from step 2000 to step 3000. BA divides this trajectory into 20 blocks, where each block represents the mean of 50 points, forming a matrix that is applied to an SVD to obtain the projection matrix $P$, as detailed in Appendix B. In contrast, TT uses only the points from step 2980 to step 3000.
>
> **W3:** The provided results should contains more detailed analysis of the metrics, for example, one can provide a similar Figure as Figure 4 from [1].
>
> **A3:** Figure 4 in [1] focused log-likelihoods on the UCI task, which we have reported in Tables 4 and 5 for comparison. Our evaluation extends beyond log-likelihood to include new metrics like Bayes factors and evidence ratios.
>
> We emphasize that BNNs are designed to evaluate uncertainty in both model weights and predictions. Metrics like test log-likelihood primarily reflect accuracy on test data, which, as we observed, can often be achieved with a single model lacking proper uncertainty quantification. To address this limitation, in Section 4.2, we proposed new quality evaluation criteria.
>
> **W4:** Incomplete comparisons between methods. For example, TT (RQMC) is missing from Tables 7 and 8. Also, there is no comparison between RQMC and SNIS.
>
> **A4:** Thank you for highlighting this. We have included TT (RQMC) results in Tables 7 and 8 in the updated manuscript. The results show that TT (RQMC) achieves similar accuracy to TT (ESS) but with lower computational cost. Furthermore, BA (RQMC) outperforms TT (RQMC) on both the CIFAR and corrupted CIFAR datasets.
>
> Table 7: Classification accuracy (ACC(\%)) on CIFAR datasets
>
> |Models|TT(RQMC)|BA(RQMC)|
> |-|-|-|
> |VGG-16 on CIFAR10|91.76±0.37|91.94±0.51|
> |PreResNet164 on CIFAR10|95.05±0.12|94.92±0.06|
> |VGG-16 on CIFAR100|68.19±0.58|68.33±0.49|
> |PreResNet164 on CIFAR100|76.82±0.19|77.30±0.35|
>
> Table 8: Classification accuracy (ACC(\%)) on corrupted CIFAR datasets using PreResNet164
>
> |Severity|TT(RQMC)|BA(RQMC)|
> |-|-|-|
> |1|94.24±0.10|94.51±0.04|
> |2|92.66±0.20|93.30±0.08|
> |3|90.52±0.27|91.29±0.12|
> |4|86.86±0.42|87.86±0.52|
> |5|69.65±1.94|72.02±1.73|
>
>
> Regarding RQMC-IS vs. SNIS, Theorem 2 demonstrates that RQMC consistently outperforms IS in convergence rate, supported by Table 2's RMSE results. Thus, we prioritized RQMC-IS over SNIS.
>
> **W5:** There is some inconsistency between the results from the paper Figure 4 and the prior work. Is there any difference in the experiment setup that caused this difference?
>
> **A5:** The key difference lies in the choice of the priors.
> We use a fixed prior independent of the subspace (Lines 166-168), ensuring a consistent posterior across the entire parameter space. In contrast, [1] modifies the prior within each subspace, causing different prior across different subspaces.
>
> A fixed prior ensures the posterior reflects only subspace properties, not artifacts from varying priors. For example, if the subspace center $\hat{w}$ shifts within the same subspace, then $\mathcal{Z}$ remains unchanged and the posterior structure should not vary. We clarified this in the revised manuscript (Lines 168-170).

---

> > ### Author Response · Authors · 2024-11-24
> > **Response Contd.**
> >
> > **W6:** RQMC-IS seems to require evaluation of $p(D|z)$ using all training data and couldn't be estimated using mini-batches like VI.
> >
> > **A6:** This is not accurate. Our RQMC-IS implementation computes sample weights (Eq.(9)) on mini-batches, followed by normalization, eliminating the need for full dataset evaluations at one time. This approach avoids the expensive gradient evaluations required by many other inference methods like variational inference.
> >
> > **W7:** What type of VI did the authors use in their paper? Have they considered comparison with SVI [2] or other scalable methods?
> >
> > **A7:** We used VI with a fully factorized Gaussian approximation, which was performed on mini-batches, following the same updating methods as used in [1].
> >
> > ---
> >
> > [1]. Izmailov et al. "Subspace inference for Bayesian deep learning", 2020.

---

> > > ### Comment · Reviewer_bE76 · 2024-11-30
> > >
> > > Thank you for the clarification. But I still have several concerns:
> > >
> > > 1. The fact that Algorithm 2 from Izmailov et al. utilizes only one point in each block instead of the average of the points needs additional analysis to substantiate the statement that "the corresponding SVD decomposition no longer aligns with the interpretations of PCA" and show importance of such alignment. I believe it is important to compare quantitatively the proposed method with Algorithm 2 with c > 1.
> > >
> > > 2. Concerning W3 I still think that the provided results cannot show a statistically significant difference between BA and TT or between RQMC-IS and ESS/VI in terms of test log-likelihood or accuracy, so statements like "Tables 4 and 5 present the test log-likelihoods for UCI-Small and UCI-Large datasets, respectively, where our method achieves higher log-likelihoods on most datasets." are inaccurate.
> > >
> > > 3. Can you clarify the computational complexity of evaluating predictive posterior on testing data? As I understand it, for $N$ samples from the posterior $z_{1}, ..., z_{N}$, a training dataset of size $|D|$, and a testing dataset of size $|d|$, it requires $\mathcal{O}(N|D|) + \mathcal{O}(N|d|)$ operations. At the same time, VI requires $\mathcal{O}(L|D|) + \mathcal{O}(N|d|)$ operations to fit posterior approximation and evaluate test samples, where $L$ is the number of training epochs. So if I understand correctly, VI can overperform RQMC in terms of computational complexity. It is hard to say which one scales better with an increase in training dataset size. In your work, you selected $L$ to be larger than $N$ (2000), and it seems to be excessive. Have you conducted experiments with a lower number of training iterations? How does the quality of VI depend on the number of training iterations?

---

> > > > ### Author Response · Authors · 2024-12-02
> > > > **Thanks for the feedback**
> > > >
> > > > Here are our replies to your comment:
> > > >
> > > > ---
> > > >
> > > > **Comments**:
> > > >
> > > > **C1:** Needs additional analysis to substantiate the statement that "the corresponding SVD decomposition no longer aligns with the interpretations of PCA".
> > > >
> > > > **A1:** Thank you for pointing this out. The statement refers to the mathematical property that, by taking only the last point in each block (as in Algorithm 2 from [1]), the resulting trajectory will deviate from the global mean. This deviation leads to SVD results that no longer correspond to the principal directions derived from a PCA decomposition of the trajectory matrix.
> > > > As a result, the interpretability of the subspace in the PCA sense is compromised.
> > > >
> > > > We acknowledge that a quantitative comparison of our method with Algorithm 2 from [1] under $c > 1$ would further substantiate this claim. However, since [1] does not provide a specific strategy for choosing $c$, we have not included such comparisons in this version of the manuscript. We recognize the value of such an analysis and will consider it for future work.
> > > >
> > > > **C2:** Statistical significance of BA and RQMC-IS results.
> > > >
> > > > **A2:** Thank you for raising this concern. Our statements refer to higher mean test log-likelihoods across most datasets; however, we acknowledge that the differences often fall within the standard deviations, which limits the statistical significance of these results. We will revise these statements to more accurately reflect the observed trends without overstating the superiority of our method.
> > > >
> > > > Furthermore, the primary contribution of the proposed metrics (Bayes factors and evidence ratios) lies in their ability to evaluate subspace quality, which goes beyond traditional performance metrics like test log-likelihood or accuracy.
> > > > As discussed in Sections 4.2 and 4.3, these metrics provide additional insights into model evidence of different subspaces, which are essential in Bayesian settings and are not captured by conventional metrics alone.
> > > >
> > > > **C3:** Clarify the computational complexity of evaluating the predictive posterior on testing data, and compare it to VI.
> > > >
> > > > **A3:** We agree that the computational complexity is $\mathcal{O}(N|D|) + \mathcal{O}(N|d|)$ for RQMC-IS, and $\mathcal{O}(L|D|) + \mathcal{O}(N|d|)$ for VI. Thus, the relative efficiency depends on the relationship between $L$ and $N$. In our experiments on synthetic regression tasks, we followed the settings in [1], using $L = 2000$ to ensure the convergence of the posterior approximation. We acknowledge that a smaller $L$ might suffice in certain cases.
> > > >
> > > > We emphasize that VI provides a variational approximation to the posterior, while MCMC and RQMC methods guarantee near-exact results when the sample size is sufficiently large. As shown in Lemma 1 and Theorem 2 of our work, RQMC-IS can converge to the true posterior predictive under some assumptions. Furthermore, RQMC-IS operates effectively in low-dimensional subspaces without requiring gradient evaluations. We will highlight this comparison in the revised manuscript.
> > > >
> > > > ---
> > > >
> > > > [1]. Izmailov et al. "Subspace inference for Bayesian deep learning", 2020.

---

### Meta-Review · Area_Chair_P9ck · 2024-12-22

**Metareview:**

This paper proposes a novel method for approximate Bayesian inference in deep learning, by marginalizing over a subspace of the model parameters.  The underlying idea is that performing Bayes over the many parameters of a deep network is intractable in general, but is made more practical by doing so over a lower-dimensional subspace.  They do this by building up the sub-space over the whole trajectory of stochastic gradient descent.  This builds on a wide literature of methods attempting to do approximate Bayesian inference on deep networks practically by marginalizing over only part of the model or a low-dim / low-rank subspace.  The authors also propose metrics to evaluate the quality of the subspace through Bayes factor and the prior-predictive.

The reviews were mixed but averaging to reject with (6, 5, 3), and all reviewers responded to the author rebuttal.  The reviewers seemed to agree that the paper is interesting, well-written and technically correct.  Bayesian deep learning is a very active sub-field and thus the topic is relevant to the community.  However, the consensus of the reviewers seemed to be that the contributions of the paper were too incremental (even the 6/Accept commented on this).   The reviewers pointed out that the method was quite similar to Izmailov et. al (TT and SWA, where the difference is mostly that the authors use the tail of the SGD trajectory rather than the whole trajectory as proposed in this paper).  The reviewers also had concerns that the experiments didn't seem to back up the claims of the authors, i.e. that the proposed improvements of the method didn't consistently seem statistically significant.

Given that the reviewers argue that the work is not quite strong enough for acceptance (novelty and experiments), and that the rebuttal didn't change their perspective, the recommendation is to reject.  The combination of an incremental contribution and not-too-strong empirical results suggests that the paper is not quite ready / strong enough for publication.  Hopefully the reviews will be useful to improve the paper for a future submission.

**Additional Comments On Reviewer Discussion:**

The authors responded to all reviews and the reviewers all acknowledged their response.  None of the reviewers changed their scores after reading the response.

Across all reviewers, the reviewers were not convinced by the response regarding the novelty of the approach in comparison to TT and SWA.  The authors argued that the difference was somewhat more significant than the reviewers' interpretation, but I would agree that the added clarification didn't seem to shed much more light on the difference.

Regarding the strength of the experimental results, the authors agreed that their results weren't always statistically significant and agreed to tone down their claims somewhat.  Of course, this limits the strength of the proposed method.

---

### Decision · Program_Chairs · 2025-01-22

Reject